# Phytochemistry, Ethnopharmacology, Pharmacokinetics and Toxicology of *Cnidium monnieri* (L.) Cusson

**DOI:** 10.3390/ijms21031006

**Published:** 2020-02-03

**Authors:** Yue Sun, Angela Wei Hong Yang, George Binh Lenon

**Affiliations:** School of Health and Biomedical Sciences, RMIT University, Melbourne 3083, Australia; s3574776@rmit.edu.au (Y.S.); angela.yang@rmit.edu.au (A.W.H.Y.)

**Keywords:** she chuang zi, cnidii fructus, herbal medicine, natural product, phytotherapy

## Abstract

*Cnidium monnieri* (L.) Cusson (CMC) is a traditional Chinese herbal medicine that has been widely grown and used in Asia. It is also known as “She chuang zi” in China (Chinese: **蛇床子**), “Jashoshi” in Japan, “Sasangia” in Korea, and “Xa sang tu” in Vietnam. This study aimed to provide an up-to-date review of its phytochemistry, ethnopharmacology, pharmacokinetics, and toxicology. All available information on CMC was collected from the Encyclopedia of Traditional Chinese Medicines, PubMed, EMBASE, ScienceDirect, Scopus, Web of Science, and China Network Knowledge Infrastructure. The updated chemical structures of the compounds are those ones without chemical ID numbers or references from the previous review. A total of 429 chemical constituents have been elucidated and 56 chemical structures have been firstly identified in CMC with traceable evidence. They can be categorized as coumarins, volatile constituents, liposoluble compounds, chromones, monoterpenoid glucosides, terpenoids, glycosides, glucides, and other compounds. CMC has demonstrated impressive potential for the management of various diseases in extensive preclinical research. Since most of the studies are overly concentrated on osthole, more research is needed to investigate other chemical constituents.

## 1. Introduction

*Cnidium monnieri* (L.) Cusson. (CMC) is the dry fruit of the Umbelliferae plant *Cnidium monnieri* (L.) from the Apiaceae family. Figure 1A shows the medicinal plant of *Cnidium monnieri* (L.). Its pharmaceutical name, English name and Chinese Pinyin name are Cnidii Fructus, Cnidium seed, and She chuang zi, respectively. As this herb is widely grown in China, Japan, Korea, and Vietnam, it is also known as “Jashoshi” in Japanese, “Sasangia” in Korean, and “Xa sang tu” in Vietnamese. In China, CMC is grown in most parts of the country. The main growing provinces are Hebei, Jiangsu, Zhejiang, Shandong, and Sichuan (Figure 1B). The first record of CMC was in Shennong’s Classic of Materia Medica (Shen Nong Ben Cao Jing). As for its properties, CMC is acrid, bitter, warm, and slightly toxic [1]. According to the latest review of CMC, 364 of its components have been identified, which mainly include coumarins such as osthole, imperatorin, bergapten, isopimpinellin, xanthotoxol, xanthotoxin, cnidimonal, cnidimarin, and glucosides. CMC is renowned for its broad range of pharmaceutical properties to treat female genitals, male impotence, frigidity, skin diseases and exerting antipruritic, anti-allergic, antidermatophytic, antibacterial, antifungal, and anti-osteoporotic effects [1]. Since the earlier review covered up to 2015, and more research projects have been conducted regarding CMC since then, it is necessary to update the relevant knowledge in a timely manner. This study thus aimed to provide an up-to-date review on the phytochemistry, ethnopharmacology, pharmacokinetics, and toxicology of CMC. 

## 2. Results

A total of 1176 studies were identified through the literature search, of which 901 studies were excluded due to duplication, or no mention of phytochemistry, pharmacology, pharmacokinetics, or toxicology. Two hundred and seventy-five studies are thus included in this review. Among them, 72 studies correspond to phytochemistry, 188 studies are on pharmacology and 12 studies are related to pharmacokinetics and toxicology, three studies did not belong to any of these three categories but feel within the scope of this study. The study selection process is illustrated in Figure 2.

### 2.1. Phytochemistry

In total 429 chemical constituents have been elucidated and 56 chemical structures (Table 1) have been revealed for the first time in CMC with traceable evidence. They can be categorized as coumarins, volatile constituents, liposoluble compounds, chromones, monoterpenoid glucosides, terpenoids, glycosides, glucides, and other compounds.

#### 2.1.1. Coumarins

Coumarins are the general term for family natural compounds which possess a benzoquinone α-pyran one core. They mainly exist in the flowers, stems, leaves, and fruits of Umbelliferae and Rutaceae plants in the free state form or in combination with sugars to form glycosides. In terms of pharmacological effects, coumarins possess anti-hypertension, anti-inflammatory, analgesic, and anticancer effects [28,29]. Coumarins include 2′-acetylangelicin, bergapten, imperatorin, osthole, isoimperatorin, isopimpinellin, xanthotoxin, columbianetin, isoporalen, fraxetin, 6,7,8-trimethoxycoumarin, murrayacarpin A, limettin, scoparone, isofraxidin, 7-methoxy-8-formylcoumarin, 2′-deoxymeranzin hydrate, oxypeucedanin hydrate, peroxymurraol, peroxy-auraptenol, auraptenol, micromarin-F, demethylauraptenol, albiflorin-3, osthenol, 7-demethyl-suberosin, cnidilin, angenomalin, meranzin hydrate, cnidimoside A, kaempferol, isorhamnetin, quercitrin, luteolin, quercetin, hesperidin, rutin, psoralen, oxypeucedanin, cnidimol D, 2′-hydrate-deoxymeranin, phlojodicarpin, (*E*/*Z*)-7-methoxy-8-(3-methylbuta1,3-dien-1-yl)-2*H*-chromen-2-one, karenin, columbianadin, cniforin A, daucosterol, xanthotol, *O*-isovalerylcolumbianetin, iisogosferol, archangelicin, (3′*R*)-3′-hydroxycolumbianetin, cniforin B, 6-methoxy-8-methylcoumafin, cnidimonal, cnidimarin, 1′-*O*-β-d-glucopyranosyl (2*R*,*3S*)-3-hydroxynodakenetin, 7-*O*-methylphellodenol-B, 7-methoxy-8-(3-methyl-2,3-epoxy-1-oxobutyl)chromen-2-one, 3′-*O*-methylvaginol, hassanone, *E*-murraol, Z-murraol, 3′-*O*-methylmurraol, Rel-(1′*S*,2′*S*)-1′-*O*-methylphlojodicarpin, (1′*S*,2′*S*)-1′-*O*-methylvaginol, and isobergapten.

#### 2.1.2. Chromones

Chromones are a family of compounds extensively distributed in Nature with multiple biological activities that include antitumor, antimicrobial, antiviral, anti-inflammatory, and antioxidant effects [30]. The isolated chromone chemical compounds are cnidimol B, peucenin, 5,7-dihydroxychromone, 5-*O*-methylvisamminol, 4′-*O*-β-d-glucosyl-5-*O*-methylvisamminol, hamaudol, 2,5-dimethyl-7-hydroxychromone, cimifugin, 5-hydroxychromone-7-*O*-β-d-glucoside, eduotin IV, oroselone, cnidimol A, cindimol F, cindimoside A, hydroxycnidimoside A, undulatoside A, saikochromoside A, cnidimoside B, 2-methyl-5-hydroxy-6-(2-butenyl-3-hydroxymethyl)-7-(β-d-glucopyranosyloxy)-4*H*-1-benzopyran-4-one, monnieriside A, monnieriside B, monnieriside C, monnieriside D, monnieriside E, monnieriside F, monnieriside G, and 6′-mydroxylangelicain.

#### 2.1.3. Triterpenoids, Glycosides, and Glucides

Triterpenoids are compounds with a carbon skeleton based on six isoprene units evolved from the acyclic C_30_ hydrocarbon squalene [31]. Glycosides are compounds where the hemiacetal or hemiketal hydroxyl groups of a monosaccharide is condensed with the hydroxyl group of a second molecule with water elimination. Cnidioside B is a representative compound in this category [32]. Glucides can be represented by glycerol-2-*O*-α-l-fucopyranoside and 6-deoxy-d-glucitol [PubChem CID: 151266] [1].

#### 2.1.4. Monoterpenoid Glucosides

The main chemical compounds in this category are 3,7-dimethyloct-1-ene-3,6,7-triol 3-*O*-β-d-glucopyranoside, (2*S*,6*S*)-3,7-dimethyloct-3(10)-ene-1,2,6,7-tetrol 2-*O*-β-d-Glucopyranoside, (4*S*)-*p*-menth-1-ene-7,8-diol 8-*O*-β-d-glucopyranoside [20], 3,7-dimethyloct-1-ene-3,6,7-triol 3-*O*-β-d-gluco-pyranoside, Enzymatic hydrolysis, (2*S*)-3,7-dimethyloct-3(10),6-diene-1,2-diol 2-*O*-β-d-gluco-pyranoside, (4*S*)-*p*-menth-1-ene-7,8-diol 8-*O*-β-d-glucopyranoside, (3′*R*)-hydroxymarmesin 4′-*O*-β-d-glucopyranoside, xanthotoxol 8-*O*-β-d-glucopyranoside, 2-methyl-5,7-dihydroxychromone 7-*O*-β-d-glucopyranoside, 3,7-dimethyloctane-1,2,6,7-tetrol, (6,7-threo)-3,7-dimethyloct-3(10)-ene-1,2,6,7,8-pentol, (6,7-erythro)-3,7-dimethyloct-3(10)-ene-1,2,6,7,8-pentol, 3,7-dimethyl-1,2,6,7-tetrahydroxy-oct-3(10)-ene,1-Triacetate, 3-methyl-1,2,3,4-tetrahydroxy-butane 2-triacetate, and 3,7-dimethyl-3β,8-dihydroxy-oct-1,6-diene 3-*O*-β-d-glucopyranoside.

#### 2.1.5. Liposoluble Compounds

Liposoluble compounds in this category can be represented by 2,4-dimethylhexane and 38 other compounds. Among these compounds, phytol (15.98%), hexadecenoic acid (12.73%), 9,12,15-octadecatrienoic acid (11.02%), 9-octadecenoic acid [PubChem CID: 637517] (6.33%) and 9,12-octadecadienoic acid (4.77%) are the main constituents of the liposoluble constituents in the fruit of CMC [33].

#### 2.1.6. Volatile Oil

In this group, according to the CMC sample from Shandong Province in China, the highest concentration substance is limonene (18.90%), followed by α-pinene (16.40%), borneyl acetate (11%), camphene (7.44%), osthol (5.24%), β-pinene (3.43%), l-borneol (1.58%), and hexadecenoic acid (1.05%) according to a GC-MS analysis method [34]. The identified compounds include α-pinene, camphene, limonene, bornyl acetate, 2-phenyl-2-(phenylmethyl)-1,3-dioxolane, cyclobutanol, pyruvic acid, 8-hexyl-8-pentylhexadecane, 1,2,4-benzenetricarboxylic acid-1,2-dimethyl ester, 2-[4-(1,1-dimethylethyl)phenoxy]-propanoic acid, 2-methyldocosane, *O*-acetylcolumbianetin, isopropyl-3-methylbutanoate, tricyclene, β-pinene, myrcene, α-terpinene, 3-7-dimethyl-1,3,7-octatriene, β-ocimene, γ-terpinene, *cis*-sabinene hydrate, terpinolene, 1,3,8-p-menthatriene, sabinene hydrate, linalool oxide, *p*-mentha-1,5,8-triene, trans-*p*-2,8-menthadien-1-ol, neo-allo-ocimene, camphor, *p*-mentha-2,8-dien-1-ol, l-borneol, 4-methyl-1-(1-methylethyl)-3-cyclohexen-1-ol, *cis*-dihydrocarvone, bicycle[5.1.0]octane,, α-fenchyl acetate, perillyl alcohol, 3,7-dimethyl-8-(1-methylethylidene)-2,6-octadienoic acid methyl ester, carvyl acetate, neryl acetate, α-copaene, β-bourbonene, β-farnesene, germacrene D, β-bisabolene, δ-cadinene, caryophyllene oxide, geranyl pentanoate, *cis*-asarone, 6,10,14-trimethyl-2-pentadecanone, 3-carene, d-limonene, 1-methyl-4-(1-methylethyl)-cyclohexene, 2,6-dimethyl-2,4,6-octatriene, 2-isopropenyl-3-methylenecyclohexanol, acetic acid 2-methyl-5-isopropenyl-2-enyl ester, neryl phenylacetate, 7,11-dimethyl-3-methylene-1,6,10-dodecatriene, eudesma-4(14),11-diene, 3-methyl-2-phenylethyl butanoic acid ester, α-farnesene, 3,7-dimethyl-2,6-octadienyl butanoic acid ester, 1,2,3,5,6,8a-hexahydro-4,7-dimethyl-1-(1-methylethyl)-naphthalene, 1,4-diethyl-1,4-dimethyl-2,5-cyclohexadiene, 1*H*-indole-3-butanoic acid, 6-isopropenyl-3-methyl-cyclohex-1-enol, 2-(4a-methyl-8-methylene-decahydro-naphthalene-2-yl)-propan-2-ol, tricyclo [4.3.1.0(3, 8)] decan-10-ol, 2,2-dimethylpropionic acid 5-isopropenyl-2-methylcyclohex-2-enyl ester, allyl phenoxyacetate, 5,5-dimethyl-8-methylene-1,2-epoxycyclooct-3-ene, 6-isopropenyl-3-methyl-cyclohex-2-enol, 2-pentacosanone, (*Z*,*Z*)-9,12-octadeca-dienoic acid, *E*-9-tetradecenoic acid, pentatriacontane, (Z)-3-heptadecen-5-yne, tetratetracontane, α-cadien, asaron, carveol, cnidiadin, diosmetin, Dl-umtatin, cnidimol C, cnidimol E, 5-formylxanthotoxol, edultin, (3′*R*)-3′-hydroxy-columbianedin, 3,7-dimethyl-3(*E*)-octene-1,2,6,7-tetraol, l-camphene, bornyl isovalerate, 2-butene, 2-methylacrolein, toluol, isobutyric acid isopropyl ester, 2-ethylidene-1,1-dimethylcyclopentane, 2-methylbutyric acid isopropyl ester, pentanoate isopropyl ester, 1,7,7-trimethylbicyclo [2.2.1] heptane, 4(10)-thujene, 1,4,8-menthatriene, *p*-cymene, 3,5-dimethylstyrene, 1,3,7-octatriene,3-7-dimethyl- 1,5,8-menthatriene, 2,2,3-trimethyl-(*R*)-3-cyclopentene-1-acetaldehyde, 1-methyl-4-(1-methyl-ethenyl)-7-oxabicyclo[4,1,0]heptane, myrtenol, dihydrocarvone, verbenone, *trans*-carveol, carvol, pinocamphone, isopiperitenone, fructus perillae aldehyde, *Z*-methyl geranate, geranyl acetate, benzyl isovalerate, phenethyl isobutyrate, decahydro-2α-methyl-6-methylene-1-(1-methylethyl) cyclobutyl [1,2,3,4] dicyclopentenyl, *trans*-caryophyllene, geranyl isobutyrate, β-eudesmol, cyclofenchene, β-terpinene, isoborneol, azulene, 1(7),8(10)-*p*-menthadien-9-ol, dipentene oxide, α-cubebene, α-bergamotene, α-elemene, dimethyl ketene, undecane, *cis*-carveol, 7*H*-furo[3,2-g] [1] benzopyran-7-one,9-[(4-hydroxy-3-methyl-2-buten 1) oxy]-[*E*], nerolidol, nerolidol isomer, *cis*-isopropenyl-2-methylene-3-cyclohexyl-acetate, propionate, *tert*-butyl phenyl acetate, diisobutyl phthalate, linoleic acid, oleic acid, stearic acid, arachidic acid, ethyl arachidate, propionic-2-methyl-1-methylethyl ester, nonane, α-methylbenzyl ethanol, *trans*-sabinene hydrate, hydroxycitronellal, 1-methyl-4-(1-methylethenyl) cyclohexene, β-terpineol, α-pinene oxide, vanillin, 4-(1-methylethyl)-benzyl alcohol, 3-cyclohexene-1-methanol, α,α,4-trimethyl-2-butyl-1-methylpyrrolidine, pentanoate-1,3,3-trimethylbicyclo [2,2,1] hept-2-yl-ester, 1-methyl-3-(1-methylethyl)phenyl-2-methyl-5-(1-methylethyl)-2-cyclohexen-1-ketone, β-dihydrofuran, 1,7,7-trimethylbicyclo[2.2,1]heptane-2-ol acetate, 14-hydroxycaryophyllene, *trans*-carvacryl acetate, α-muurolene, cubenol, *trans*-dihydro thuja alcohol, bicyclogermacrene, dihydrocalamenene, 6-methyl-α-ionone, *trans*-sabinene sesquihydrate, widdrol, 4-α-hydroxydihydro-furan-agar, α-bisabolol, 2-camphanyl angelic acid ester, methyl jasmonate, furfuryl heptadecenone, *cis*,*cis*-farnesol, 3-methyl-2-butene-1-ol, 2-pentyl-furan, dehydro-*p*-cymene, nonanal, β-linalool, fenchol, 2,6-dimethyl-1,3,5,7-octatetraene, α-terpinenol, thymol methyl ether, undecanal, ylangene, β-elemene, dodecanal, isocaryophyllene, α-cedrene, α-santalene, thujopsene, α-caryophyllene, acoradiene, β-chamigrene, γ-muurolene, 2-tridecanone, β-himachalene, ar-curcumene, γ-elemene, α-cadinene, α-longipinene, α-himachalene, germacrene B, perillyl acetate, geranyl propionate, geranyl 3-methylbutyrate, myristic acid, hexahydrofarnesyl acetate, 3,7,11,15-tetramethyl-2-hexadecen-1-ol, palmitoleic acid, 2-methyl-2-β-butane-1-ol, isopropylisobutyric acid, α-sung terpene, 2(10)-sung terpene, pentylbenzene, 1,3,3-trimethylbicyclo [2.2.1] heptan-2-ol 2-acetate, 5-propene-2-cyclopenten-1-ol, 2-methyl-5-propene-2-cyclopentene-1-acetate, 1-phenyl-1-pentanone, 1-propyl-3,4-dimethoxybenzene, α-selinene, cinnamyl acetate, cedrol, 8-(3-methyl-2-β) hem, *p*-mentha-e-2,8(9)-dien-1-ol, 3-cyclohexaen-1-ol, 4-methyl-l-(1-methylethyl)-3-cyclohexaen-1-ol, octadecanal, 1-octadecene, (3β,24*S*)-stigmast-5-en-3-ol, 6,6-dimethyl-2-bicyclo [3,1,1] heptane, 4-methyl-l-(1-methylethyl)-2-cyclohexene-1-ol, 5-isopropenyl-2-methyl-cyclohex-2-enyl propionic acid ester, 1,4-dimethyl-3-cyclohexene-1-ethanol, tricyclo [4.4.0.0 (2,8)] decan-4-ol, and 5-isopropenyl-3-methyl-cyclohex-1-enol.

#### 2.1.7. Liposoluble Compounds

The chemical compounds in this category include 2,4-dimethylhexane, 2,3-dimethylhexane, 2,3,4-trimethylhexane, 2-methylheptane, 3-methylheptane, octane, 5-ethyl-2,4-dimethyl-2-heptene, 1,2,5,5-tetramethyl-1,3-cyclopentadiene, heptadecane, *Z*-11-tetradecenoic acid, octadecane, pentadecanoic acid, nonadecane, (*Z*)-7-hexadecenoic acid, eicosane, heptadecanoic acid, 6,9,12-octadecatrienoic acid, 9,12-octadecadienoic acid, 9,12,15-octadecatrienoic acid, phytol, 7,10,13-eicosatrienoic acid, heneicosane, docosanoic acid, tricosanoic acid, tetracosanoic acid, nonacosane, stigmast-4-en-3-one, hexacosanoic acid, tocopherols, and octacosanoic acid.

#### 2.1.8. Trace Elements

CMC contains various trace elements, including Cu, Fe, Zn, Mn, Sr, Ca, and Mg. Among these elements, the concentration of Fe can be as high as 136.69 mg/kg, followed by Zn and Cu at 51.72 mg/kg and 33.49 mg/kg, respectively [35].

#### 2.1.9. Other Compounds

In addition to the above categories, there are other compounds such as β-sitosterol, *p*-coumaric acid, etc. and a few benzofuran compounds such as palmitic acid, sitosterol, *p*-coumaric acid, thymine, hypoxanthine, uracil, l-(+)-valine, d-phenylalanine, cnideoside A, and cnideoside B.

In sum, all the chemical constituents isolated from the herb are summarized in Appendix A. Compared to the previous review [1], 80 more chemical compounds have been identified in the current review and 56 of them with their chemical structures have been reported in Table 1.

### 2.2. Pharmacology

#### 2.2.1. Nervous System and Mental Illness

##### Memory Loss and Neuron Degeneration

The main early symptom of Alzheimer’s disease (AD) is learning deterioration and memory loss. The animal experiments have indicated that osthole could significantly enhance the synaptic transmission function of rat hippocampal dentate gyrus in a dose-dependent manner at different dosages of 5, 15 and 30 ng. Therefore, osthole could improve learning and memory ability [36]. Another study showed osthole (40 mg/kg) remarkably improved memory acquisition and consolidation of obstacles and direction to identify obstacles, and prolonged the time from decapitation to hypoxia, but had no significant improvement in memory reproduction disorders. The mechanism could be attributed to inhibition of mRNA expression of inflammation-related genes such as Tnf-α, II-1β, Nos2, and COX-2 in rat hippocampus [37]. Another study revealed that osthole (40 mg/kg) could suppress NF-κB activation, and reduce the protein expression of IL-1β, and TNF-α [38]. Osthole may attenuate spatial learning and memory loss and neuronal ultrastructure injury induced by β-amyloid protein fragment 25-35 (Aβ_25-35_). In terms of the neuronal ultrastructure injury alleviation, osthole displayed a similar effect as donepezil which is the first-line medication for AD management [39]. It had also been found that osthole (12.5 mg/kg and 25 mg/kg) regulated the expression of apoptosis-related proteins Bcl-2 and Bax in hippocampus of AD rats, increased the ratio of Bcl-2/Bax, and had anti-apoptotic and protective effects on the hippocampal neurons, which may have a role in improving learning and memory impairment. The Bcl-2 gene family is an important apoptosis-regulating gene. The family has many members, among which Bcl-2, Bcl-xl, and Bcl-w have anti-apoptotic effects, and Bax, Bak, and Bok have apoptosis-promoting effects. The mechanism of apoptosis is related to the protein expression of apoptotic genes. Bcl-2 and Bax proteins are a pair of antagonistic proteins, apoptosis is accelerated by redundant Bax while apoptosis is inhibited by excessive Bcl-2 protein [40]. Osthole may also take a counter-AD effect by inhibiting cyclic nucleotide phosphodiesterases (PDEs) which are responsible for the degradation of cAMP and cGMP [41]. Furthermore, the PDE4 and PDE5 inhibition was the key to an anti-neurotoxicity effect caused by Aβ25-35 with Osthole at 40 mg/kg [42]. From in vitro studies, it is also found that osthole (50 µmol/L and 100 µmol/L) promoted the proliferation of neural stem cells cultured in vitro and its mechanism could be associated with the activation of Notch 1 gene and Hes 1 gene in Notch signaling pathway by increasing the expression of Notch 1 gene and Hes 1 gene [43]. Another in vitro experiment demonstrated that the protective effect of osthole on astrocytes was related to its concentration. The low concentration (0.1 μmol/L and 0.5 μmol/L) of osthole had a better protective effect on astrocytes. The mechanism is that osthole inhibits Aβ-induced IκBα hyperphosphorylation and attenuates NF-κB over-activated expression in astrocytes to exert a protective effect and improve neuronal cells’ ability to resist Aβ_25-35_ [44]. Another study demonstrated that osthole (50 µmol/L) had the best protective effect on SH-SY5Y cells transfected with APP595/596 gene by inhibiting the mRNA and protein expression of BACE1 [45]. Osthole (50 µmol/L) could inhibit CAMKK2/AMPK signal pathway to reverse the neuronal synapse structure lesion induced by Aβ [46]. Osthole (100 µmol/L) exerted anti-apoptotic/pro-proliferative effects by activating Wnt/β-catenin signaling and stimulating neural stem cell [47]. Osthole (100 µM) increased bone marrow-derived-neural stem cell viability and decrease apoptosis on H_2_O_2_ induced oxidative injury. It also regulated the expression of apoptotic genes and prevented bone marrow-derived-neural stem cells apoptosis mediated by H_2_O_2_ through modulation of PI3K/Akt-1 signaling pathway [48]. Osthole (50 µmol/L and 100 µmol/L) had also been proven to take neural protective effect by underlining mechanisms, enhancing neural stem cell survival via the miR-9 signaling pathway, and promoting the differentiation of neural stem cells into neurons by upregulating miR-9 [49]. Osthole (100 µmol/L) could downregulate the expression of neuro stem cells p16, and upregulate the expression of CDKD1 and PRB protein in the result of neuro stem cells proliferation and differentiation [50].

##### Brain Hemorrhage

Osthole (15 mg/kg) enhanced superoxide dismutase level in brain tissue and dipped interleukin-8 level in serum to exerting protective effects on swelling brain tissue via in vivo study [51].

##### Epilepsy

From an in vivo study, osthole (50 mg/kg) protected hippocampal neurons in kainite by inhibiting the activity of caspase-3 and caspase-9 protein [52]. Another possible mechanism was to downregulate the expression of Puma protein in hippocampal neurons [53]. Osthole (50 mg/kg) protected neurons in the hippocampus of epileptic rats via augmenting the expression of Kv1.2 in hippocampus CA_3_ neurons [54]. From the in vitro study, osthole (10 mmol/L) also protected HT22 cells from glutamate excitotoxicity via the activation of the PI3K/Akt signaling pathway [55].

##### Hypnosis and Sedation

In in vivo studies, different concentrations of CMC extract (62.5 mg/kg), 50% to 95% alcohol extract could exert hypnotic and sedative activity. Specifically, 50% alcohol extract could exert an utmost sedative effect, while 95% alcohol extract could exert an utmost hypnotic effect [56]. The incubation period of each group of *Cnidium* was not significantly different from that of diazepam, which could induce sleep rapidly and significantly prolong the duration of sleep, and the hangover response and tolerance in adverse reactions were weaker than that of diazepam [57]. Another study showed the hypnotic active component of CMC (130–520 mg/kg) exerted a hypnotic effect on animal models but had no influence on the animals’ learning and memory functions [58]. For hypnotic activity alone, the chemical ingredients (from CMC (25, 50, and 100 mg/kg)) which include total coumarin, osthole, imperatorin, isopimpinellin, and bergapten can boost the expression of Cry1, Per1, and Per2 gene via inhibiting the hypnosis of hippocampal clock and Bmall gene expression. It achieved the hypnotic effect by increasing the expression of the inhibitory neurotransmitter γ-aminobutyric acid and decreasing the expression of excitatory neurotransmitter glutamic acid [59]. After osthole intervention (25 mg/kg per day for 7 days), the malondialdehyde levels in both hippocampus tissue and serum were decreased. Additionally, the superoxide dismutase activity in the hippocampus became normal [60]. The hypnotic efficacy, spatial cognition and spatial reference memory preservation function by CMC (40, 80, and 160 mg/kg) have been verified in another study with a mouse model [61].

##### Anxiety

Osthole (1.75–14 mg/kg) was found to have an anxiolytic effect in an in vivo study by suppressing anxious behavior after irritation [62].

#### 2.2.2. Immune System

##### Allergy

From in vivo studies, for anti-asthmatic activity, osthole (25, 50, 100 mg/kg) kept the airway inflammation at bay, the production of Th2 cytokines and OVA-specific IgE, the recruitment of eosinophils and mucus overproduction [63]. Osthole (10 mg/kg and 50 mg/kg) improved goblet cell proliferation and reduced fibrosis in lung tissue and mucus secretion. Besides, it is also found that osthole could interfere with the production of white blood cells and eosinophils [64]. Osthole (10 and 40 mg/kg) reduced airway mucus secretion by downregulating mCLCA3, mRNA and MUC5AC protein levels, and suppressing IL-4 induced eotaxin in BEAS-2B cells through inhibition of STAT6 expression [65,66]. Osthole (200 and 500 mg/kg) exhibited an inhibitory effect for contact dermatitis rat model especially for the symptom of ear swelling [67]. From in vitro studies, osthole (150, 300, 450 ng/mL) acted as a natural histamine antagonist that targeted TNF-α, IL-6 and IL-10 cytokine secretion levels without a toxic effect for peripheral blood mononuclear cell. Thereby, this phenomenon could be the evidence for the beneficial role of osthole in allergic conditions [68]. For allergic skin diseases, osthole (0.25, 0.5, 1 g/mL) exerted the anti-allergic effect via inhibiting the inflow of Ca^2+^ channel in mast cell and reducing the release of histamine [69]. Osthole affected peripheral blood mononuclear cells’ cytokine secretion to the same extent as the anti-histamine medication, fexofenadine. Specifically, osthole (150 ng/mL) was more effective than fexofenadine in terms of IL-13 blocking ability. The blockage effect on other cytokines such as IL-1β, IL-10, and TNF-α was similar to fexofenadine [70]. The extract of CMC (100 µg/mL) had immune regulatory functions by inhibiting retinoic acid receptor α reporter gene transcription which could affect the immune cell production and ratios [71].

#### 2.2.3. Circulatory System

##### Atherosclerosis

Through in vivo studies, osthole (10 mg/kg, and 30 mg/kg for 16 days) downregulated NF-κB and TLR4 expression to reduce inflammatory factors such as TNF-α and IL-1β, and restrained intimal hyperplasia and vascular smooth muscle cell proliferation [72,73]. It is observed that when the estradiol 2 level in serum was in a certain amount, osthole (10–20 mg/kg and 10 mg/kg) treated hyperlipidemia and an ovariectomized model could exhibit lower levels of lipoprotein lipase, hepatic lipase, very low-density lipoprotein, chylomicron, and triglyceride [74]. From in vitro studies, osthole (20, 40 µmol/L) suppressed the apoptosis of human umbilical vein endothelial cells with the mechanism of activating Akt/Enos/NO signal pathway and inhibited atherosclerotic formation by regulating lipid and countering inflammation [75].

##### Cardiac Diseases

Through docking analysis and chromatographic retention time measurements, imperatorin exerted the same effect as verapamil which is a calcium antagonist for hypertension and cardiovascular disease management [76]. From in vivo studies, total coumarins as a family of compounds remarkably reduced the area of myocardial infarction and enhance the cardiac function after the infarction. Meanwhile, at 20 mg/kg it also decreased the number of mitochondrial fragments and improved the expression of dynamin-related peptide-1 and optic atrophy-1 which were associated with mitochondrial morphology [77]. Another study indicated that osthole (50 mg/kg) could protect myocardial ischemia-reperfusion injury from mitochondrial mediating apoptosis via suppressing the activation of Wnt/-catenin/p53 signal pathway [78]. Osthole (25 mg/kg) exerted a similar effect by taking antiapoptotic effects via PI3K/AKT signaling pathway [79]. It (25 mg/kg) also resisted myocardial apoptosis by upregulating the expression of Bcl-2 protein and downregulating the expression of Bax protein, therefore to increase the ratio of Bcl2/Bax [80]. For right ventricle remodeling, osthole (10 and 20 mg/kg) adjusted the right ventricle remodeling induced by monocrotaline. The mechanism could be associated with the up-regulation of the expression of peroxisome proliferator-activated receptor α and peroxisome proliferator-activated receptor γ [81]. Pulmonary artery remodeling could also be reversed by osthole at (10 and 20 mg/kg) via augment expression of p53 and lessen the expression of proliferating cell nuclear antigen and Ki67 [82]. From in vitro studies, for anti-arrhythmia activity, osthole with a concentration from 100–500 µmol/L could dramatically inhibit sodium current of ventricular myocytes in the animal model [83]. Osthole (IC_50_ = 162.1 56.2 and 300 µmol/L) could also inhibit L-type (Cav1.2 and Cav1.3) and N-type (Cav2.2) calcium channels to restrain the calcium current, therefore to shorten the duration of cardiac cell potential [84]. Osthole could inhibit the potassium channels to attenuate the electric current of cardiocytes. Specifically, the outward potassium channel could be reversed when the IC_50_ is around 101.1 µmol/L, the inward potassium channel can be influenced when the concentration reaches 200 µmol/L [85]. Osthole (10^−9^–10^−5^ mol/L) not only dilated the artery ring of normal human and rat lung tissue but also dilated the pulmonary arteries on both species. The mechanism could be related to either calcium release from the sarcoplasmic reticulum or ion channel [86]. Osthole (5 µg/mL) managed myocardial fibrosis by inhibiting collagen I and II expression and diminished their ratio via the transforming growth factor-β (TGF-β)/Smad signaling pathway in TGF- β1 overexpressed cardiac fibroblasts (CFs) [76].

##### Hypertension

Imperatorin, xanthotoxol, and other imperatorin derivatives can relax the mesenteric artery, basilar artery, and renal artery. It is also found that two derivatives showed the promising results as a vasodilator agent for the mesenteric artery. They are 9-(2-(diisopropylamino) ethoxy)-*7H*-furo[3,2-g] chromen-7-one and 9-(2-(pyrrolidin-1-yl) ethoxy)-*7H*-furo[3,2-g] chromen-7-one. Another pair of derivatives have shown the potential to be a vasodilator agent for basilar artery [87]. From in vivo studies, Imperatorin (25 mg/kg) exerted influences directly on the cardiac muscle. In detail, imperatorin inhibited cardiac myocyte protein synthesis induced by angiotensin II, it also attenuated pathological myocardial hypertrophy and pathological cardiac fibrosis. Besides these, it prevented the transition to heart failure [88]. Osthole inhibited the elevation of systolic blood pressure in stroke-prone spontaneously hypertensive rats. It also induced a remarkable increase in hepatic 3-hydroxy-3-methylglutaryl coenzyme A reductase mRNA expression induced the cholesterol decreased in the liver [89]. For imperatorin, after 10 weeks of intervention (6.25, 12.5, and 25 mg/kg per day), the middle and high dose group demonstrated a significant reduction of mean blood pressure on renal induced hypertension rat models. Besides, the kidney function parameters were attenuated by these two dose groups. Specifically, there was a decreasing trend in both angiotensin II levels and plasma endothelin levels while an increasing trend occurred in plasma nitric oxide synthase (NOS) activity and nitric oxide (NO) levels. Imperatorin could also reduce renal excretion of 8-iso-prostaglandin F2α in renal induced hypertension rat models with no proteinuria increase detection. Xanthine oxidase as a major supply of renal reactive oxygen species was remarkedly decreased after the imperatorin intervention. As for mRNA expression reductions had been noticed which includes p22phox, p67phox, p47phox, and gp91phox, the protein levels of gp91phox and p47phox were diminished in renal cortical tissue [90]. It is also found that imperatorin (6.25, 12.5, and 25 mg/kg) could relax aortic rings. For calmodulins’ interactions in molecular docking, imperatorin had the same binding site as verapamil (a commonly used hypotensive medication) [91]. As tested by both in vitro studies, Imperatorin (1–100 μm) could restrict angiotensin II-induced cardiac myocyte protein production, modulate pathological myocardial hypertrophy, pathological cardiac fibrosis, and prevent heart failure [92]. Osthole (1–100 µmol/L) triggered an endothelium-independent relaxation in mice aortic rings via blocking Ca^2+^ channel to achieve a vasorelaxation effect [93].

##### Organ Ischemia

For renal ischemia protection, osthole (40 mg/kg) could lower several biomarkers, including creatinine and blood urea nitrogen, and it was observed via histological evidence that renal tubular dilatation, luminal obstruction, necrosis, shedding, and renal interstitial inflammatory cell infiltration were markedly reduced. It was also found that osthole reduced the production of reactive oxygen species which could induce the injury on mitochondrial when its production and metabolism were imbalanced. Osthole enhanced the ATPase activity in mitochondrial. It also suppressed the activation of the apoptotic signal pathway and the expression of apoptosis-related proteins and inhibited the expression of cytochrome C [94]. In another study, osthole played an important role in inflammation and oxidative stress during the progression of renal ischemic injury. In details, osthole reduced the expression of proinflammatory cells such as tumor necrosis factor-α, monocyte chemotactic protein-1, and interleukin 6 (IL-6) mRNA at a dosage of 20 mg/kg. Osthole also dramatically lowered the expression of oxidative stress enzyme, malondialdehyde. In addition, it could enhance the bioactivity of catalase, glutathione peroxidase, and superoxide dismutase which is the enzyme against oxidative stress [95,96]. Osthole (20, 40, and 80 mg/kg) exerted similar effects on other organs such as retina and intestinal. Taking the retina as an example, in addition to the above mechanisms, osthole could elevate the amplitude of electron retinol graph-b [97]. For intestinal ischemia-reperfusion lung injury model, osthole (5 and 25 mg/kg) could significantly improve the lesions of lung tissue induced by oxidative stress and apoptosis and also induce the reduction of Caspase-3 expression [98]. In 50 mg/kg osthole intervention group, the survival rate was impressive, as high as 85%, and the oxygenation and mean arterial pressure was also improved in both 10 mg/kg and 50 mg/kg group, the pathologic examination revealed the improvement in lung injury in both dosage groups, reactive oxygen species, lung wet-to-dry weight ratio, pulmonary permeability index was dramatically decreased in both dosage groups, the malondialdehyde level was decreased while superoxide dismutase activity was increased in both dosage groups, inflammatory cytokine levels such as IL-6 and TNF-α were reduced in both dosage groups, myeloperoxidase as a neutrophil indicator enzyme was lowered by osthole in both dosage groups [99]. In the heart transplant-induced injury model, the biomarkers of heart function including lactate dehydrogenase, cardiac creatine kinase, and creatine kinase-muscle/brain were down-regulated by osthole at 25 mg/kg. For the microstructure of myocardium after osthole intervention, the sarcoplasmic reticulum was slightly dilated, the myofilament was dissolved, the nuclear membrane was intact, the mitochondria were swollen, but the morphology was not severely disordered and vacuolar degeneration was rare, without dissolution [100]. For cerebral ischemia, osthole exerted an anti-cerebral ischemia effect by suppressing mitochondria-mediated apoptosis via taking below actions. It improved the neurological symptoms and brain swelling at best dosage of 100 mg/kg, inhibited reactive oxygen species (ROS) production, enhanced the membrane potential and the adenosine triphosphate (ATP) vitality of mitochondrial, suppressed apoptosis-inducing factor and cytochrome C migration, attenuated apoptosis-related signal pathways. Specifically, upregulated caspase 3, caspase 9 and Bcl-2/Bax protein expression, downregulated cleavage-caspase 3 and cleavage-caspase 9 protein expression. Thus, osthole’s anti-cerebral ischemia effect could be associated with ROS and ATP level drop, the membrane potential stability during the perfusion period of mitochondrial, and mitochondrial membrane permeability transition pore from opening [101]. It is noted that osthole (4, 8, and 16 mg/kg) could elevate the level of superoxide dismutase and glutathione while decreases the level of malondialdehyde [102]. It is also found that osthole (25, 50, and 100 mg/kg) could impede cerebral infarction, hippocampus neuronal lesion, and apoptosis created by middle cerebral artery occlusion/reperfusion (MCAO/R) model via initiating Notch 1 signaling pathway in a dose-dependent manner in vivo. The similar findings were obtained by the oxygen-glucose deficiency/reperfusion (OGD/R) model intervened by primary neurons in vitro [103]. When it comes to the protein expression, matrix metalloproteinase-9 (MMP-9) was detected to be downregulated in ischemia/reperfused brain with osthole at 100 mg/kg [104]. The learning and memory impairment were ameliorated by osthole (12.5 and 25 mg/kg) after cerebral ischemia-reperfusion injury. Osthole lifted hippocampal long-term potentiation level in the model rats via attenuating glutamic acid and γ-aminobutyric acid levels [105]. By inhibiting IL-8 and IL-1β, the vitality of myeloperoxidase could be impeded, reducing the infiltration of neutrophils in brain tissue and local tissue inflammation, enhancing Na^+^, k^+^-ATPase, Ca^2+^-ATPase activity. To alleviate nitric oxide (NO) induced injury to neuron, osthole (5 and 10 mg/kg) inhibited inducible nitric oxide synthase (iNOS) vitality and NO level [106,107].

##### Thrombosis

From in vivo study, osthole (10, 20, and 40 mg/kg) exhibited a remarkable anti-clot effect on the venous and artery-vein bypass of rats. The mechanism could be attributed to NO and 6-keto-prostaglandin 1α (6-Keto-PGF1α) level augment thromboxane B2 (TXB2) level decrease and TXB2/6-Keto-PGF1α ratio [108]. For the in vitro study, osthole (0.95–3.78 mg/mL) inhibited the rising calcium concentration in platelet induced by thrombin by restraining the expression of Inositol triphosphate receptor-3 [109].

#### 2.2.4. Symptoms and States of Undefined Origin

##### Pain

Post-surgery administration of osthole (50 µL) in an early stage can mitigate the nucleus pulposus-induced radicular inflammatory pain via inhibiting the expression of phosphorylation extracellular signal-regulated kinase and cyclooxygenase-2 mRNA in the spinal dorsal horn and nitric oxide synthase in dorsal root ganglion [110,111]. In another study, osthole (50 µL) could upregulate calcitonin gene-related peptide receptors 1 (CGRPR1) to easing sciatica pain by downregulating CGRPR1 protein expression [112]. From in vitro studies, osthole (0.2, 0.5, and 1 g/mL) could block the conduction of the action potential of nerve in pain signal transduction [113]. Osthole (20 and 50 g/L) could also inhibit the acid-stimulated change of dorsal root ganglion neuron’s membrane potential of rats with nucleus pulposus-induced hyperalgesia [114].

##### Hepatic Steatosis

Osthole (10, 20, and 40 mg/kg) reduced hepatic oxidative stress in the treatment of alcoholic fatty liver with 20 mg/kg providing the best outcome [115]. Osthole (10, 20, and 40 mg/kg) was effective against fatty liver by suppressing hepatic sterol regulatory element-binding protein-1c/2 mRNA expressions and modulating the sterol regulatory element-binding protein-1c/2-mediated target gene expression such as fatty acid synthase, cholesterol 7α-hydroxylase, and low-density lipoprotein [116]. Osthole (10, 20, and 40 mg/kg) reduced the accumulation of lipids in the liver in the mechanism of antioxidation and TNF-α production suppression [117]. From in vitro studies, osthole (100 µmol/L and 25–200 mg/mL) could reduce triglycerides and free fatty acid in hepatocytes with the mechanism of peroxisome proliferator-activated receptor activation, sterol regulatory element-binding protein-1/2 reduction, fatty acid synthase, and diacylglycerol acyltransferase gene expressions and increment of carnitinepalmitoyl transferase, fatty acid synthase 4, liver fatty acid, cholesterol 7α-hydroxylase, and 3-hydroxy-3-methylglutaryl-CoA binding protein gene expressions [118,119].

##### Inflammation

From in vivo studies, osthole (50 mg/kg) could benefit inflammatory bowel disease by increasing IL-10 and reducing the levels of TNF-α and IL-17. When it came to the combination with sulfasalazine, the results showed a significant reduction in the levels of TNF-α, IL-17, and IFN-γ, meanwhile, it was also found a remarkable increase of IL-4 and IL-10. Thus, osthole could reverse the polarized balance of Th1, Th2, and Th17 cells by reduction of proinflammatory cytokines and increase of anti-inflammatory cytokines [120]. Inflammation played an important role in the brain injury model. Osthole (10, 20, and 30 mg/kg) could restrain the expressions of inflammatory factors and decrease the apoptosis of neurons [121]. Specifically, osthole could lower the ratio of Bax/Bcl-2 and the expression of caspase-3 mRNA at the dosage of 20 and 30 mg/kg. Therefore, it could achieve inhibition for nerve cell apoptosis caused by brain injury. It could also reduce the positive cells of caspase-3 when the dosage is at 30 mg/kg [122]. Cnidium lactone micro-emulsion (2, 4, and 8 mg/mL) for local external application remarkably reduced auricular swelling induced by xylene and vola pedis swelling induced by egg white [123]. Peroxyauraptenol and auraptenol as chemical compounds of CMC could exert an anti-inflammatory effect. Peroxyauraptenol was 10 times stronger than auraptenol in terms of the inhibition of IL-6 secretion and NO production. For iNOS and pro IL-1β expression inhibition, peroxyauraptenol was also more efficient than auraptenol. For peroxyauraptenol alone (10 μM), it could also lower the phosphorylation levels of mitogen-activated protein kinases and PKC-α/δ, reduce NLRP3 inflammasome activation by mitigating mitochondrial damage, inhibit IL-1β precursor expression [124]. It was observed in animal experiments that osthole was associated with the inflammatory responses which included inflammatory cell infiltration, erythrocyte exudation and capillary congestion. It (40 mg/kg) inhibited the classic signal pathway of JAK2/STAT3 and expression of inflammatory cytokines such as IL-6, IL-1β, and TNF-α [125]. Osthole (100 and 200 mg/kg) demonstrated protective effects for sepsis-induced acute lung lesions via suppressing the inflammatory reaction, oxidative stress, and cell apoptosis [126]. From in vitro studies, osthole (4, 7 and 10 µg/mL) could shield inflammatory BV2 cells from inflammation-induced by lipopolysaccharide stimulation in the mechanism of inhibition of NF-κB and Nrf2 signaling pathways [127]. Osthole (12.5–100 µmol/L) could dramatically suppress the production of NO, IL-6 and TNF-α and other inflammatory cytokines. Moreover, it also inhibited the protein expression of iNOS and COX-2 [128]. The reason behind osthole’s (10 μg/mL) suppression of TNF-α, NO and COX-2 expression could be partially attributed to inhibition of PKC-α, PKC-ε, JNK1/2, p38, ROS, and NF-κB pathways [129]. Osthole (IC_50_ value of 0.005 µg/mL) and cnidimol A (IC_50_ value of 3.2 µg/mL) could be used as promising candidates against formyl-l-methionyl-l-leucyl-l-phenylalanine-induced O_2_ generation and elastase release for the management of multiple inflammatory diseases [130].

##### Fatigue

Through in vivo studies, for fatigue management, the traditional Japanese formula “Zena F-III” (10 mL/kg) which contains CMC and other 14 herbs had been vindicated to be able to block the calcium to take its therapeutic effect. The target chemical compound in this study was Osthole [131]. In another study, CMC (0.75–4.5 g/kg per day for 3 days) could mitigate the repercussions of high-intensity exercise on serum testosterone by maintaining it at normal physiological levels. It could also increase hemoglobin and glycogen reserves and promote protein synthesis and inhibit the degradation of amino acid and protein [132].

##### Oxidation

From an in vitro study, for antioxidant effect, osthole (0.1–0.6 mL) had scavenging capability against three types of free radicals, including O_2_^−^, OH and 1,1-diphenyl-2-picrylhydrazyl-based radical [133].

#### 2.2.5. Other Infectious and Parasitic Diseases

##### Bacterial Infectious Disease

The in vitro experiments showed the extract of CMC could significantly inhibit *Escherichia coli* in a dosage-dependent manner. The minimal inhibitory concentration was 250 mg/mL, and the minimum bactericidal concentration was 500 mg/mL [134]. Osthole (IC_50_ = 34 µM) and imperatorin (IC_50_ = 28 µM) could inhibit *Staphylococcus aureus*. Osthole demonstrated 4-fold inhibition in the minimal inhibitory concentration of ciprofloxacin which is one of the frontline anti-bacteria medications [135]. CMC extract inhibited urea plasma urealyticum, and the minimal inhibitory concentration ranged from 62.5 to 125 μg [136]. The combinations of matrine or oxymatrine with CMC (60 mg/mL) took their synergistic inhibitory effect *against individual microorganisms such as Candida albicans, Staphylococcus aureus, and Escherichia* coli [137]. The co-extracted volatile oil (100 µL) formed by both CMC (5 g/mL) and *Zanthoxylum bungeanum* (5 g/mL) showed a better anti-vaginitis effect than a single application which indicated the synergistic effect of these herbs [138]. Both in vivo and in vitro studies showed CMC (300 mg/kg per day for 5 days) enhanced macrophage phagocytosis, increased the survival rate on mice after *Escherichia coli* infection and protected the host from infection via improving bacterial reproduction in the peripheral blood. Besides, the inflammatory cytokines had gone through significant changes on Raw 264.7 cells after CMC intervention (10, 30, and 100 µg/mL). IL-12, IFN-γ, and TNF-α levels in the serum dramatically increased while IL-6 level remarkedly decreased [139].

##### Virual Infectious Disease

Osthole had anti-HIV activity by inhibiting the export of viral regulatory protein, viral regulatory protein from the nucleus to the cytoplasm with IC_50_ value at 1.6 μm [140].

##### Fungal Infectious Disease

From in vitro studies, CMC inhibited intestinal *Candida* species like *C. albicans*, *C. krusei*, and *C. tropicalis* at an average minimum inhibitory concentration of 12.5 g/L [141]. Osthole (4–16 µg/mL) had a remarkable synergistic effect with fluconazole against fluconazole-resistant *C. albicans*. The possible mechanism could be related to endogenous reactive oxygen species augmentation [142]. Xanthotoxin as a common methoxylated furanocoumarin had been reported to inhibiting the human fungi *Candida albicans* and *Cryptococcus laurentii* [143]. Osthole (0.5 g) with the integrated lactonic ring had the anti-fungal activity to all three tested funguses including *T. mentagrophytes*, *T. purpureatum*, and *Microsporum gypseum*. When the lactonic ring was broken, the antifungal effect did not exist on *T. mentagrophytes* [144].

##### Parasitic Disease

From in vitro studies, osthole (120 µm) inhibited the formation of blood vessels against echinococcosis without the high risk of liver and kidney toxicity [145]. In another study, osthole (10 mg/mL) impaired the ultrastructure of *Giardia lamblia* against its infection. Specifically, the impaired suction membrane led to the unstable suction and malnutrition for *Giardia lamblia* [146]. Imperatorin (LD_50_ = 3.14 mg/L) and osthole (LD_50_ = 13.11 mg/L) could be utilized as larvicides in the management of mosquito populations, especially to target insecticide-resistant mosquito larvae [147]. Osthole (1.12, 2.24, and 4.48 mg/mL) could eliminate *Trichomonas vaginalis* with a minimal concentration of 1.12 mg/mL in the tube [148]. With 120 µM, the inhibition effect of osthole was so strong on *E. granulosus* protoscoleces that all the protoscoleces were eliminated within three days. In an in vivo study, osthole could shrink the size of metacestodes-infected mice tissue by remarkably reducing the small vesicles and blood vessels. It was also noted that the level of IL-4 and the percentage of eosinophils had been increased in the osthole-treated group. For addressing cytotoxic concerns, there were no adverse changes detected from morphological observation and liver and kidney function indexes [145]. For another kind of parasite, *Giardia lamblia*, osthole may inflict a series of structure alterations which included cell membrane damage, nuclear deformity, and sucking structure damage. Therefore, it is reasonable to deduce that all these structure alterations induced abnormal physiological functions including unstable sucking and deteriorating nutrition absorption, which led to autophagy and death [146]. The most efficient concentration was 5.0 mg/mL in the study which could approximately reduce the number of the parasite down to 21.38% with IC_50_ was 1.345 mg/mL, it also deactivated the vitality of the parasite down to less than 50% after 5 h intervention [149]. For *Trichomonas vaginalis*, it was observed that, after the osthole intervention, the body of the parasite was turned into a round shape, bubbles emerged inside the nucleus, the activity of flagellum and the undulating membrane was dramatically lessened and gradually ceased which led to the parasite body breakage. The active concentration started from 1.12 mg/mL which could ensure the 100% elimination of the parasite after the 24-h intervention [148].

#### 2.2.6. Endocrine-Metabolic System

##### Diabetes Mellitus

From in vivo studies, osthole (20 mg/kg per day for 14 days) reduced the up-regulation of the P2X_4_ receptor, and downregulated the IL-1β, TNF-α, brain-derived neurotrophic factor, and p-p38MAPK and upregulated IL-10 in diabetes mellitus [150]. For diabetes-associated cognitive decline, osthole protected neuro functions via inhibiting PI3K/Akt signaling pathway with the dosage at 50 mg/kg and exerted a similar effect to the positive control medication Donepezil [151]. From in vitro studies, osthole (5, 10, 20, and 40 µmol/L) decreased the apoptotic rate in NIT-1 cells and the expression of apoptotic proteins such as Bax and Caspase 3, increased the expression of anti-apoptotic protein Bcl-2 to achieve inhibiting cell apoptosis. The possible mechanism could be related to PPARα and endoplasmic reticulum stress [152]. Osthole (100 nmol/L) protected human umbilical vein endothelial cells from the injury induced by high glucose that could benefit the prevention and control of diabetes angiopathies [153]. Osthole (0–200 μm) activated glucose uptake but initiated deactivation in L929 fibroblast cells and inhibited uptake in HCLE cells [154].

##### Hormone Modulation

Osthole (2.5 g) and total coumarins of CMC enhanced the pituitary-thyroid axis function of the Kidney-Yang deficiency rat model by increasing the secretion and synthesis of the thyroid hormone of T3, reverse T3, and T4 [155]. For estrogen-like effects, osthole (0.75–24 mg/kg) increased the expression of uterine estrogen receptor β and surge the level of estradiol 2 in the serum [156].

#### 2.2.7. Skin and Subcutaneous Cell Tissue

##### Hypertrophic Scar Fibroblasts

From in vitro studies, osthole could inhibit the growth of hypertrophic scar fibroblasts via apoptosis and decreased the expression of TGF-β_1_ with IC_50_ value 15.5+/−2.2 µmol/L toward hypertrophic scar fibroblasts [157]. In another study, osthole could exert its maximum inhibitory effect on mice embryonic fibroblasts which are similar to human dermal fibroblasts with a dosage at 2.5 × 10^4^ mol/L [158]. Osthole (5–50 µmol/L) significantly inhibited the growth of human hypertrophic scar fibroblasts and reduced the expression of TGF-β1 [159].

##### Psoriatic Effect

From in vitro studies, osthole could induce HaCaT cells apoptosis in a dose-dependent manner which indicated osthole could exert anti-psoriatic effect by inhibiting excessive proliferation of epidermal cells and inducing apoptosis of epidermal cells. The IC_50_ was 1.75 × 10^−4^ mol/L, 1.47 × 10^−4^ mol/L, and 1.35 × 10^−4^ mol/L for 24 h, 48 h, and 72 h incubation, respectively [160]. Another study tested the CMC on the same cells with IC_50_ 114.6 µg/mL by 3-(4,5-dimethylthiazol-2-yl)-2,5-diphenyl tetrazolium bromide (MTT) assay [161].

##### Atopic Dermatitis

From in vivo *and* in vitro studies, for atopic dermatitis management, osthole (10, 20, and 40 mg/kg, 1.0 and 5.0 mg/mL) suppressed the proliferation and degranulation of sensitized mast cells and inhibited the expression of STAT5 gene and protein [162,163]. From an in vitro study, osthole microemulsion (2, 4, and 8 mg/mL) could counter histamine phosphate and 4-aminopridine (4-AP) induced pruritis by inhibiting delayed hypersensitivity caused by dinitrochlorobenzene (DNCB) which indicated the anti-pruritic effect through anti-histamine and anti-allergic reactions [164].

##### Itchiness

From the in vitro study, osthole (2, 4, and 8 mg/mL) could exert a localized anesthetic effect and inhibition effect on N calcium channel Cav2 genus II which mainly distribute on the sensory neuron’s cell body and nerve endings [164]. For therapeutic effect constituent identification, the volatile oil had the best antipruritic effect, and alcohol extract had the weakest effect [165]. From an in vivo study, osthole and isopimpinellin from CMC extract (200 and 500 mg/kg) showed an inhibitory effect on compound 48/80 caused scratching behavior [166].

#### 2.2.8. Musculature and Skeleton

##### Osteoporosis

From the in vivo studies, among CMC coumarins, six (at 5 g/kg) have shown androgen-like effects on male rats in experiments. Specifically, osthole, imperatorin, bergapten, isopimpinellin, xanthotoxin, and xanthotoxol can increase bone formation and decrease bone resorption which counters the pathogenic effects induced by prednisone. These findings shed the light for the synergic application of both glucocorticoid and coumarin extract of CMC against osteoporosis [167]. Total coumarins (2.5 mL/kg) could dramatically improve the bone density, increase IGF-1 and 25-OH vitamin D level [168]. For in vitro study, osthole and imperatorin (0.1–10 µmol/L) exerted estrogen-like effects to boost osteoblastic activity in vitro cell culture experiments [169].

As for osthole, in female rats, osthole (6.7 mg/kg) displayed an anti-osteoporosis effect on ovariectomized samples which simulated the menopause-related osteoporosis [170]. It was also revealed that osthole could increase the levels of calcium and phosphorus in serum and boost the phosphorus level in femurs. In addition, the bone density in the lumbar spine and femur showed an increasing trend, the average width and area percentage of trabecular bone were remarkedly elevated, the biomechanical index was attenuated by osthole. In detail, ultimate strength displayed an increasing trend and the bending section coefficient was significantly reduced [171]. Osthole (20 mg/kg) intervention significantly inhibited tricalcium phosphate-induced endoplasmic reticulum stress response and reduced glucose-regulated protein 78 and CAAT/enhancer-binding protein homologous protein expression, suggesting that osthole’s inhibition of tricalcium phosphate particle-induced osteolysis around the prosthesis may be achieved by regulating endoplasmic reticulum stress response [172]. The bone morphogenetic protein was enhanced by cyclic adenosine monophosphate/cAMP response element-binding protein signaling pathway which targeted the transcription factor osterix. The expression of osterix was up-regulated by osthole at 20 mg/kg [173]. From in vitro studies, osthole (1 × 10^−6^ mol/L and 1 × 10^−8^ mol/L) upregulated osteoprotegerin (OPG) mRNA expression level but did not influence on receptor activator of NF-κB ligand (RANKL) mRNA expression level. Finally, it affected the ratio of OPG/RANKL to inhibit bone resorption from osteoclasts [174]. Osthole promoted bone marrow stromal stem cells differentiation to osteogenesis at the dosage of 1 × 10^−5^ mol/L, and also increased the gene expression of bFGF, OSX, Runt-related gene 2, and IGF-1 [175]. Osthole at a concentration of 1 × 10^−5^ mol/L could stimulate differentiation and maturation of rat calvarial osteoblasts and exert no cytotoxic effect [175]. Osthole (0, 6.25, 12.5, 25, and 50 µmol/L) could inhibit the expression of proliferating cell nuclear antigen and cyclin D1 to achieve chondrocytes inhibition in a dose-dependent manner [176]. Osthole has poor water solubility, low bioavailability, and poor infiltration for bone tissue, therefore osthole could promote osteoblasts by applying a water-soluble chitosan derivative as a carrier to boost the efficacy [177]. The Wnt/β-catenin signaling pathway could be one of the mechanisms [178]. It was also found that the expression level of tartrate-resistant acid phosphatase, macrophage colony-stimulating factor, and cathepsin K mRNA could be involved in osthole activity at 1 × 10^−5^ mol/L [179]. Endoplasmic reticulum stress may be involved in the release of inflammatory mediators around the prosthesis, osteoclast genesis, and osteolysis. NFATc1 is a gene involved in the mechanism of osthole inhibition in this regard with tested dosage at 10^−6^ and 10^−5^ mol/L [180]. Bone morphogenetic protein 2 activations triggered bone morphogenetic protein signaling which was closely associated with the bone function [181]. The chemical structure of osthole could be the reason for its role in bone formation and bone resorption. It was found that osthole, icariin, and 8-prenylnaringein sharing a similar structure, namely the prenyl group. Icariin and 8-prenylnaringein have been viewed with the function of mediating bone reconditioning and prophylaxis against osteoporosis [182].

For valine, valine as a CMC chemical compound branched-chain amino acid which could maintain and stabilize skeletal muscle. The downregulation of valine could counter the prednisolone-induced osteoporosis via regulating amino acid metabolism [183].

##### Bone Fracture

For genes involvement, collagen type X, alkaline phosphatase, osteocalcin, and bone morphogenetic proteins were upregulated after osthole intervention in chondrocyte differentiation, chondrocyte maturation, and endochondral ossification [184]. From an in vivo study, total flavonoids (5 mL/kg per day for 90 days) could lower the bone fragility, curb the tendency of bone fracture and increase the biomechanical index such as the maximum load, fracture load, elastic load, elasticity, bending energy. Among them, elastic load, elasticity, bending energy was statistically significant (*p* < 0.05) [185].

##### Osteolysis

From an in vivo study, tricalcium phosphate particle-induced osteolysis could be prevented by osthole local administration (10 mg/kg) via the endoplasmic reticulum stress signaling pathway [186].

##### Periodontitis

From the in vitro study, osthole (10^−5^ m/L) with vitamin C co-treatment could increase cell sheet formation and osteogenic protein/gene expression in both periodontal ligament stem cells and jaw bone marrow mesenchymal stem cells. Osthole motivated collagen type-I, fibronectin, and integrin β1 production and calcium accumulation for extracellular matrix mineralization because of the increase of mRNA expression of alkaline phosphatase, runt-related transcription factor 2, and osteocalcin. Osthole can directly improve bone-forming activity via stimulating cell osteogenic differentiation and bone marker gene expression [187].

#### 2.2.9. Tumor Diseases

##### Digestive System Cancer

Osthole could significantly inhibit the growth of cholangiocarcinoma QBC939 cells in a dose and time-dependent manner with IC_50_ values of 0.16963 mmol/L and 0.10019 mmol/L. Besides, osthole could also influence the cell growth circle to induce apoptosis at the G2/M phase via the caspase signaling pathway [188]. Osthole suppressed the clonogenicity of esophageal cell carcinomas (ESCC) cells such as KYSE150 and KYSE410 cells in a dose-dependent manner, plus, it was also noticed that osthole’s inhibitory effect against ESCC cells was related to the build-up of cell cycle progression at G2/M phase and apoptosis induction with the activation of caspase-dependent pathway, the P13K/AKT (p-AKT) signaling pathway through the activation of PTEN which was responsible for P13K/AKT pathway negative regulation and cell growth inhibition [189]. For pancreatic cancer, osthole could inhibit pancreatic cancer progression, reduce tumor weight, suppress proliferation, migration, and cause apoptosis of pancreatic 02 cells. It was noticed that osthole prevented the infiltration of M2 macrophages in cancer and reduced the numbers of M2 macrophages in the spleen, In addition, osthole impeded the polarization of primary bone marrow cells into M2 macrophages, which was associated with the downregulation of the STAT6 and C/EBP β signaling pathways in the IL-4 induced RAW 264.7 cells [190]. In the case of gastric cancer, osthole started inhibiting BGC-823 cells at a dose of 20 µg/mL. This pattern was in a dose-dependent manner. When the inhibitory concentration (IC_50_) was 100 µg/mL, the protein expressions including caspase-3 and caspase-9 were upregulated by osthole to induce cell apoptosis [191]. Osthole (100, 150, and 200 µm) could downregulate the protein expression of MMP-9 and vimentin to suppress TGF-β dependent tumor invasion [192]. Another mechanism for anti-liver cancer activity was associated with bio-behavioral characteristics of hepatocellular cancer and vascular characteristics [193]. Osthole (0.004, 0.02, 0.1, and 0.5 µmol/L) induced apoptosis-related to the downregulation of anti-apoptotic Bcl-2 expression, upregulation of proapoptotic Bax and p53 proteins. Collectively, the activation of caspases and mitochondria in HepG2 cells was involved in this aspect [194]. When it comes to lower digestive tract cancers such as colon cancer, CMC (120–200 µg/mL) showed cytotoxic effects and induced apoptosis in HCT116 colon cancer cells, it also attenuated mitochondria-related apoptotic proteins which included Bcl-2-associated X protein (Bax) and Bcl-2-homologous antagonist killer (Bak) in a dose-dependent manner [195]. From in vivo studies, for liver cancer, osthole (0.25, 0.5 and 1.0 mmol/kg) limited the tumor growth in a dose-dependent manner without immune toxicity in mice model. The intervention of osthole discriminately increased the proportion and quantity of CD8^+^ T cells without effects on other cells in the spleen of the test model. It also stimulated the activation of CD4^+^ and CD8^+^ T cells for which infiltrated the tumor [196]. Another study showed that osthole (0.25, 0.5, and 1.0 mmol/kg) could at least partially suppress NF-κB activity to exert cancer cell inhibition and apoptosis effect [197]. NBM-T-BMX-OS01 (BMX) (20 mg/kg) is an osthole derivative which suppresses angiogenesis via vascular endothelial growth factor receptor 2 (VEGFR2) signaling and cancer cell inhibition in HCT116 colorectal cancer cell line [198].

##### Blood Cancer

From in vitro studies, osthole (100 μmol /L) demonstrated an inhibitory effect on HL-60 acute myeloid leukemia cells. It could enhance the apoptosis induced by tumor necrosis factor-related apoptosis-inducing ligand (TRAIL). It attenuated the protein expressions by descending Bcl-2 mRNA and elevating Bax and Dr5 mRNA expression. In comparison, when osthole and TRAIL were applied together, the attenuation was dramatically amplified. Osthole also improved the vitality of caspase-3, caspase-8, and caspase-9 remarkably [199]. For multiple myeloma, except downregulating Bcl-2 expression, the expression of p53 and apoptosis-related protein such as cleaved-PARP were upregulated with the tested dosage of osthole from 0 to 200 µmol/L [200]. The activation of the PI3K/AKT signaling pathway was involved in K562 leukemia cells inhibition and apoptosis promotion with Osthole from 0 to 20 µg/mL [201]. In addition to osthole (IC_50_ = 14.9 and 9.3 µg/mL), imperatorin (IC_50_ = 18.8 and 20.2 µg/mL), bergapten, isopimpinellin, and xanthotoxin all exhibited inhibitory effects on two types of leukemia cells including HL-60 and P-388 in 24 h. More importantly, osthole and imperatorin showed high sensitivity to these two types of cells and induced apoptosis in HL-60 cells [202].

##### Breast Cancer

From in vitro studies, osthole (20 µg/mL) inhibited human breast cancer cells in a dose and time-dependent manner. It exhibited a suppressive effect on tumor angiogenic factors including vascular endothelial growth factor (VEGF) and MMP-9. It could prevent the downstream of the MAPK signaling pathway via suppressing the activity of breast cancer cells HER-2/neu [203]. Osthole (20 µmol/L) could dramatically increase the sensitivity of breast cancer stem cells to TRAIL. The mechanism could be attributed to the apoptotic protease activating factor-1 (Apaf-1) upregulation. This upregulation could further promote activation between caspase-9 and Apaf-1 with the assistance of cytochrome C which initiated the process of apoptosis [204]. For breast cancer MCF-7 cell, osthole (0–100 µmol/L) induced the apoptosis of MCF-7 through activation of p53 signaling. Moreover, osthole increased the expression of Bax, p21, and Cytc and decreases Bcl-2 expression. G1 phase arrest may be one of the reasons for MCF-7 cells inhibition [205,206]. Another mechanism elaborated on the initiation of apoptosis for breast cancer cells BT-20 is the formation of RIP1-FADD-caspase-8 complex via CIAP2/RIP1 pathway [207]. The inhibition of TGF-β1 expression was against the bone metastasis of breast cancer [208].

##### Prostate Cancer

From in vitro studies, osthole (50 µmol/L) could downregulate silent information regulator 1 (SIRT1) expression in prostate cancer LNCaP cells and enhance the sensitivity to doxorubicin (DOX). Osthole enhanced the DOX efficacy on the tumor cells by p53 activation, the STRT1/p53 pathway promoted apoptosis in tumor cells from mitochondria level [209]. In another type of cell, osthole (30, 60, 90, and 120 µmol/L) showed a similar effect on DU145 cells [210].

##### Nasopharyngeal Cancer

From in vitro studies, osthole (150 mg/L) could increase sensitivity for the radiotherapy through the mechanism of downregulation of gene expression of VEGF and hypoxia-inducible factor-1α (HIF-1α) and inhibition of angiogenesis [211]. In the case of tumor stem cell proliferation, osthole (20, 40, and 80 µmol/L) exhibited a similar effect on CNE2 cells in terms of its inhibition and sensitivity for radiation therapy. The possible mechanism was related to the protein expression suppression of p-GSK-3, β-catenin, and cyclin D1 [212].

##### Cervical Cancer

From in vitro studies, it was found that, after osthole intervention (10 µg/mL, 150 µg/mL, and 200 µmol/L) on HeLa cervical cancer cells, the expression level of Bax m-RNA and Fas was elevated significantly, while the expression level of Bcl-2 mRNA was reduced dramatically. The cellular ROS level surged in a dose-dependent manner. Immune escape responsible factor human transforming growth factor β1 (TGF-β1) secretion was restricted by osthole. These findings indicated the possible mechanism was associated with the imbalance of Bcl-2/Bax which triggered apoptosis [213,214]. In another type of tumor cells, similar findings were obtained for Bax and Bcl-2 attenuation. At the concentrations of 20, 40 and 80 µg/mL, the expression levels of Cle-caspase-3 and Cle-caspase-9 were upregulated [215].

##### Melanoma

From in vitro studies, after osthole intervention (20, 40, and 80 mol/L) on melanoma A375 cells, the mRNA level of Bcl-2 declined, same as the expression of p-NF-κB and Bcl-2 protein level in a dose-dependent manner, the mRNA level of Bax and the expression of IκB-α increased. These findings were associated with suppression of NF-κB signal transduction pathway activation [216]. The melanoma B16F10 cells proliferation inhibition may attribute to methoxy group at the C-5 or C-8 position for the chemical compounds bergapten and xanthotoxin, the mechanism behind the activity of these two compounds was due to G2/M arrest via an elevation of checkpoint 1 kinase (Chk1) phosphorylation and a plummet in the level of cdc2 phosphorylation [217].

##### Urinary Cancer

From in vitro studies, for adrenocortical Y1 cancer cells, osthole (1–200 µmol/L) could attenuate adrenal cortex function and facilitate corticosterone synthesis and production via enhancing gene expressions of steroidogenic enzymes [218]. After osthole intervention (10, 25, and 50 µm), the corticosterone production was promoted with or without dibutyryl-CAMP (Bu_2_cAMP), the expression level of StAR and CYP11B1 increased in a dose-dependent manner, NR4A1, NR4A2, and NR5A1 gene expression was enhanced, the cholesterol uptake and transport to the inner mitochondrial membrane was promoted which is responsible for the synthesis of steroid hormones [219]. For bladder T24 cancer cells, osthole (50 mmol/L) could suppress the expression of COX-2, VEGF, and NF-κB, while in contrast, caspase-3 activity had been strengthened to achieve bladder cancer cell T24 inhibition [220]. Osthole (5–160 µmol/L) could reverse the T24/ADM cell drug resistance to chemotherapy drugs such as adriamycin (ADM), fluorouracil, and cisplatin. The mechanism could be related to P-gp expression downregulation [221].

##### Lung Cancer

From in vitro studies, the combined application of both osthole (50, 100, and 200 µmol/L) and cisplatin could significantly inhibit NCI-H460 cells. The mechanism was associated with the increasing production of ROS which caused the apoptosis for the tumor cells [222]. The similar inhibition phenomenon had occurred on NCI-H520 cells and LA795 cells with the dosage of osthole from 6.25 µmol/L to 100 µmol/L and from 0 µmol/L to 160 µmol/L, respectively [223,224].

##### Brain Cancer

From in vitro studies, Osthole (100 and 200 µm) restrained the proliferation and promotes the apoptosis of human glioma cells through the upregulation of microRNA-16 and downregulation of matrix metalloproteinases-9 [225]. Osthole (1, 10, and 30 µm) suppressed the migratory ability of human glioblastoma multiforme cells through inhibition of focal adhesion kinase-mediated matrix metalloproteinase-13 expression [226]. In another study, osthole (50–150 µm) exerted a remarkable inhibitory effect for the proliferation of human glioma U251 cells via the PI3K/Akt signaling pathway [227].

#### 2.2.10. Toxicity

For hepatotoxicity, firstly, osthole mitigated tamoxifen-induced liver injury by reducing the liver function parameters such as aminotransferase and aspartate aminotransferase in serum. Besides, the histological structure can be maintained by osthole pre-treatment. Secondly, it ameliorated tamoxifen-induced oxidative stress by decreasing the levels of hydrogen peroxide and malondialdehyde in serum, hydrogen peroxide and glutathione levels in liver, reactive oxygen species, and lipid peroxide levels. Another mechanism is that osthole (100 mg/kg) could dramatically inhibit the mRNA levels of cytokines including TNF-α, IL-1β, IL-6, and MCP-1, additionally, it could down-regulate the gene expressions of the CYP2D6, CYP2E1, CTP3A11, and CYP4A10 while up-regulating the gene expressions of UGT1A1, UGT1A6, UGT2B1, SULT2A1, and GSTM. It also exerted the hepaprotective effect by reducing p38 phosphorylation and exerted antioxidative effects [228]. Osthole (10 mg/kg) decreased the liver production of α-smooth muscle actin (α-SMA), inflammatory cytokines and chemokines in thioacetamide treated rats. In addition, osthole could also lessen both TNF-α-induced NF-κB activity and TGF-β-induced α-SMA in hepatic stellate cells (HSCs). Taken together, osthole could inhibit both HSC activation and liver inflammation [229].

For cardiotoxicity induced by adriamycin, after osthole intervention (20 mg/kg), the lesions on the microstructure of myocardium were mitigated, the concentration of calcium in cardiomyocytes was lowered, the bioactivity of Na^+^-K^+^-ATP enzyme on the membrane of cardiomyocytes was lessened. Therefore, the mechanism was associated with Na^+^-K^+^-ATP enzyme activation, Na^+^/Ca^2+^ exchange, calcium concentration reduction, and calcium overload avoidance [230].

For neurotoxicity, osthole (0.01, 0.05, 0.1 mmol/L) protected PC12 cells from neurotoxicity induced by 1-methyl-4-phenylpyridinium ion in a dose-dependent manner. It also indicated the mechanism behind this phenomenon is down-regulating mitochondrial cytochrome C production to achieve energy metabolism adjustment and inhibition of dopaminergic neurons apoptosis [231].

#### 2.2.11. Respiratory System

For idiopathic pulmonary fibrosis, osthole (1, 10, and 50 µmol/L) restrained its proliferation, collagen synthesis, and phenotypic differentiation. The mechanism could relate to the inhibition of the expression of the Ras C3 botox substrate1and the production of reactive oxygen species [232].

#### 2.2.12. Genito-Urinary System

For focal segmental glomerulosclerosis management, osthole (30 mg/kg) ameliorated urine protein levels, renal function, and renal lesions. It also elevated nuclear factor E2-related factor 2 protein levels, cytosolic protein levels of heme oxygenase 1, and glutathione peroxidase activity, reduced the production of reactive oxygen species and serum levels of prostaglandin E_2_, hindered renal macrophage infiltration, nuclear translocation of NF-κB, and cyclooxygenase-2 expression, prevented podocyte injury and renal apoptosis [233]. The pharmacological effects and activities associated with this herb are presented in Appendix B (Table A1).

### 2.3. Pharmacokinetics and Toxicology

The pharmacokinetic studies have been mostly concentrated on osthole and other few compounds in a series of research projects. It has been found that osthole is widely distributed and absorbed in the plasma but slowly eliminated from the body. More importantly, it could possibly infiltrate the blood-brain barrier and the blood-testis barrier. It densely distributes on the liver, kidney, brain, and spleen, and its residues stay longer in the fatty tissues including epididymis, testicular tissues, and lungs [234]. Other experiments have revealed similar findings in both mice and rabbits [235,236]. For osthole metabolism and excretion, in phase I, the main metabolic pathway is 7-demethylation, 8-dehydrogenation, hydroxylation on coumarin feature and 3,4-epoxide formation, while in phase II, the sulfated conjugate is the main metabolite of osthole. CYP3A4 could be the metabolic enzyme responsable for osthole’s elimination in vitro. Troleandomycin inhibited osthole metabolism in a concentration-dependent manner between 0–200 μmol/L. After elimination, osthole could be found as the original form in the rat urine [237]. The experiment performed on three critical chemical constituents bergapten, imperatorin, and osthole revealed the pharmacokinetic parameters T max were 2.18 ± 0.58 h, 2.03 ± 0.34 h, 2.75 ± 0.24 h, respectively, the C max readings were 1.18 ± 0.22 µg/mL, 7.03 ± 1.27 µg/mL, 13.16 ± 1.37 µg/mL, respectively [238]. The recommended dosage is 30–120 mg/kg of osthole safe application [239]. It is indicated that osthole eliminations can be affected by some other factors and pre-existing disease in the host. For example, the rising body temperature can prolong the absorption for osthole. Ultimately it can increase the value of the area under the curve (AUC). Likewise, the acute renal failure can lead to the surge of osthole in plasma and extension of its elimination. Therefore, CMC dosage should be adjusted accordingly in the clinical practice to avoid toxic accumulations especially for the patients with such factors and diseases [234].

For toxicology, according to the records in Chinese Pharmacopoeia (2015 Edition), CMC has a limited degree of toxicity. Minor adverse effects have been reported after its use, including bitter mouth, drowsiness and stomach discomfort. The majority of CMC toxicity was transient and reversible [1]. The liver and renal toxicity have been reported from animal experiments on different experimental species such as zebrafish and mice. Accordingly, multiple toxic experiments have been conducted to investigate the mechanisms. The findings revealed that, for liver toxicity, osthole could exert toxic effects on the L02 liver cells in the time and dosage-dependent manner with the IC_50_ at 290.30 µmol/L in 24 h. The mechanism could be related to L02 cells apoptosis via the mitochondria pathway. Meanwhile, osthole was also able to inhibit cell reproduction through down-regulating p-Histon H3 (ser10) expression [240].

From the comprehensive acute and long-term toxic experiments, Cnidium alcohol extract in all three dosage groups (9.00, 4.50, 2.25 g/kg) demonstrated liver toxicity effects on mice in terms of the liver function biomarkers including aspartate aminotransferase, alanine aminotransferase, and glutamic transpeptidase. All three biomarkers’ readings were lower than the normal control group after the cnidium alcohol extract intake. The blood biochemistry readings were also under the influence; however, it is not in a dose-dependent manner. More importantly, the medium lethal dosage (LD_50_) was gauged as 17.45 g/kg by gavage which is equal to 116 times of clinical dosage in the study [241]. In contrast, the LD_50_ number in another study was as low as 3.45 mg/kg [242]. In another in vivo study, the histopathological examination showed the cnidium extract intervened group with hepatic eosinophilic degeneration and blood vessels surrounding inflammatory cell infiltration in liver cells. For renal toxicity, the cnidium extract intervened group demonstrated blood urea nitrogen and serum creatinine ratio elevation which indicates its adverse effects on the kidney function [243]. In another aspect, in the case of medication combined application, osthole could increase the possibility of adverse effects from other medications by inhibiting the CYP3A4 vitality both in vitro and in vivo. Due to the fact CYP450 plays a significant role in medication interactions, for the total CYP450 system, CYP3A4 accounts for 30% in the liver and 70% in the intestinal wall. Therefore, the inhibition effect on CYP3A4 was detrimental to the safety of multiple medication intake [244]. Intriguingly, the liver toxicity could be mitigated by combined application with another herb, *Glycyrrhiza uralensis* [245].

## 3. Discussion and Conclusions

This review has summarized the knowledge of chemistry including chemical compound names and chemical structures of the constituents without references from PubChem, ChemSpider references, and previous reviews. Four hundred and twenty-nine (429) chemical constituents have been elucidated and 56 chemical structures have been summarized here for the first time, either in the form of PubChem or ChemSpider ID numbers or in the form of the details of the chemical structures. The traceable evidence distinguishes this research from previous ones which claimed the chemical compounds without evidence of the chemical ID numbers and the chemical structures. It is critical to present the chemical compounds with traceable evidence to prevent duplications and errors since a large number of chemical compounds have multiple names or synonyms. For the identified chemical compounds, they could be categorized into different derivatives: coumarins, which are the most dominant group, volatile constituents, liposoluble compounds, chromones, monoterpenoid glucosides, terpenoids, glycosides, glucide, and other compounds. In terms of pharmacological activities, osthole has showcased comprehensive activities including memory and learning enhancement, anthelmintic activity, anti-allergic activity, anti-atherosclerosis activity, analgesic activity, antibacterial activity, anti-cardiac diseases, anti-diabetic activity, anti-epilepsy activity, anti-fatty liver activity, antifungal activity, antiglioblastoma activity, hypnotic and sedative activity, anti-hypertension activity, anti-hypertrophic scar activity, organ ischemia-reperfusion, anti-cerebral ischemia activity, anti-inflammatory activity, bone health, anti-parasite activity, skin conditions, anti-thrombosis activity, anti-cancer activity, anti-hepatotoxicity activity, and other activities. From in vivo and in vitro studies, Imperatorin demonstrated anti-hypertension effects, anti-osteoporosis effects, anthelmintic effects, and anti-bacterial effects, and from computational analysis, an anti-cardiovascular disease effect had been uncovered. Total coumarins may exert anti-myocardial infarction effects, hypnotic effect, and anti-osteoporosis effect. Total flavonoids could display anti-osteoporosis effects. For toxicology, the current studies were mainly focused on osthole. Although the findings have presented multiple concerns regarding organ toxicity and other adverse effects. The induced dosages were still controversial due to its significant data disparity. For pharmacokinetics, the identified studies were all focused on osthole metabolism in different animal models. The findings present that osthole is widely distributed and absorbed in a fast manner in the physiological system, ranging from most of the major organs to fatty tissues after oral administration. Therefore, it raised the concern that the absorption rate is not satisfactory for clinical applications, especially in phase I. However, this shortcoming could be partially mitigated by using different administration routes and different forms of pharmaceutical products in the market such as gels and microemulsions [246].

In summary, CMC has demonstrated impressive potential for the management of various diseases in extensive research studies. However, most of such studies are overly concentrated on osthole, therefore, more research is needed to investigate other chemical constituents in this herb, and to examine the bioavailability of CMC, more pharmaceutical studies are needed.

## 4. Materials and Methods

The literature was searched in the following databases from their respective inception until May 2019: Encyclopedia of Traditional Chinese Medicine [247], PubMed (https://www.ncbi.nlm.nih.gov/pubmed/), EMBASE (https://www.embase.com/), ScienceDirect (https://www.sciencedirect.com/), SCOPUS (https://www.scopus.com/freelookup/form/author.uri), Web of Science (https://mjl.clarivate.com/), China Network Knowledge Infrastructure (http://new.oversea.cnki.net/index/). The keywords used for the literature search included: Chinese name (蛇床子), Common English name (Common Cnidium Fruit), botanical or scientific name (*Cnidium monnieri (L.)* Cusson), and pharmaceutical name (Cnidii Fructus). The selection criteria included process controls of the herbal substances, reporting reference standards such as authentication of reference materials and profile chromatograms, and analytical procedures and validation data. Papers in English or Chinese language were included in this review. Scientific rigidity was determined by the chemical markers of herbs by strict parameters in testing, quantitative, and qualitative measures of the bioactive components, such as high-performance liquid chromatography, fingerprint spectrum, correlations differentiation, and stability evaluation, reference standards, and toxicological assessments. Plant voucher specimens were a guarantee for traceability of the plant material and data verification for other researchers or commercial purposes [248]. The chemical formulas of the compounds of CMC were acquired from selected studies. The chemical constituents with PubChem (https://pubchem.ncbi.nlm.nih.gov/) or ChemSpider (http://www.chemspider.com/) reference or with references from the previous comprehensive review were spared to draw chemical structures, the chemical constituents without these references were summarized. Chemical structures and molecular names were obtained using ChemDraw Professional 170.

## Figures and Tables

**Figure 1 ijms-21-01006-f001:**
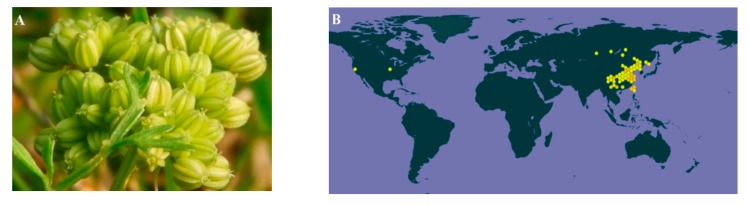
(**A**) The medicinal plant of *Cnidium monnieri* (L.)*,* created by Penny Wang and published by iNaturalist (Record license http://creativeco...censes/by-nc/4.0/). (**B**) the global distributions of *Cnidium monnieri* (L.) Cusson (https://www.gbif.org/species/3034720). CMC is mainly grown in China, Japan, Korea, and Vietnam and scarcely grown in America and the Russian Far East.

**Figure 2 ijms-21-01006-f002:**
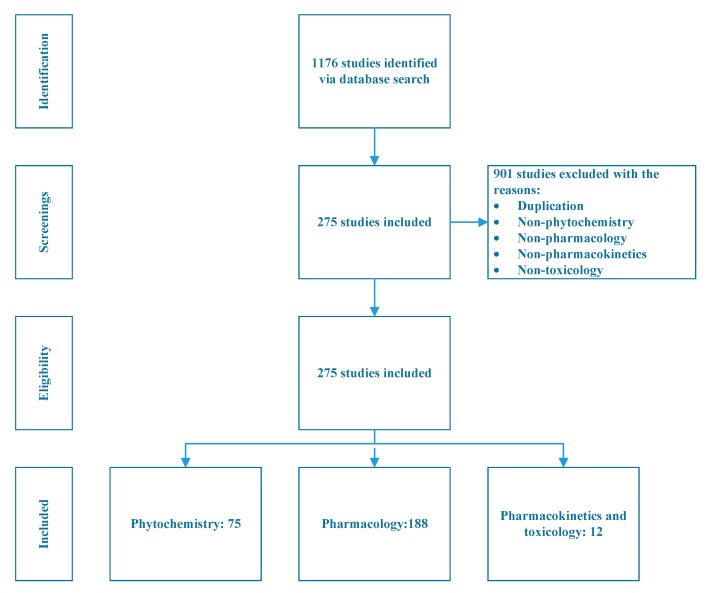
Study selection process for included studies related to *Cnidium monnieri* (L.) Cusson.

**Table 1 ijms-21-01006-t001:** Molecular formula and chemical structures of compounds derived from *Cnidium monnieri* (L.) Cusson (80 in total, 56 with chemical structures).

No.	Compound Derivatives	Compound	Chemical Formula	Chemical Structure	References
1	Coumarins	(*E*/*Z*) 7-methoxy-8-(3-methylbuta1,3-dien-1-yl)-2*H*-chromen-2-one	C_15_H_14_O_3_	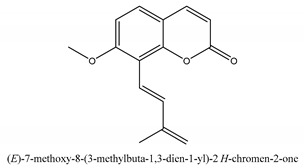	[2]
2	(*Z*,*Z*)-9,12-octadeca-dienoic acid	C_18_H_32_O_2_	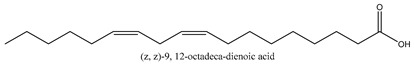	[1]
3	1,2,3,5,6,8a-Hexahydro-4,7-dimethyl-1-(1-methylethyl)-naphthalene	C_15_H_24_	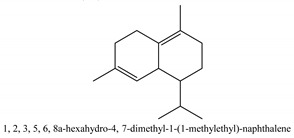	[3]
4	1,4-Diethyl-1,4-dimethyl-2,5-cyclohexadiene	C_12_H_20_	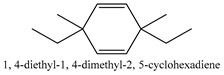	[3]
5	1,4-Dimethyl-3-cyclohexene-1-ethanol	C_10_H_18_O	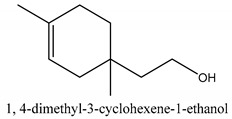	[3]
6		1,7,7-Trimethyl-bicyclo [2.2,1] heptane-2-ol acetate	C_12_H_20_O_2_	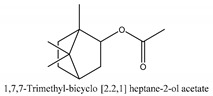	[4]
7	1-Methyl-3-(1-methylethyl)-phenyl	C_10_H_14_	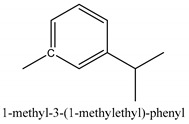	[4]
8	1-Propyl-3,4-dimethoxybenzene	C_11_H_16_O_2_	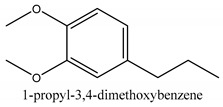	[5]
9	2(10)-Sung terpene	N/A	N/A	[5]
10	2-(4a-methyl-8-Methylene-decahydro-naphthalene-2-yl)-propan-2-ol	C_15_H_26_O	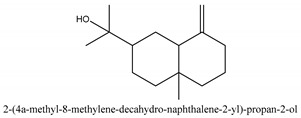	[3]
11	2,2-Dimethyl-propionic acid 5-isopropenyl-2-methylcyclohex-2-enyl ester	C_15_H_24_O_2_	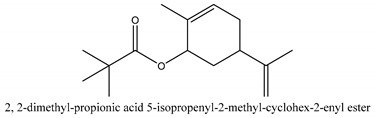	[3]
12	2-[4-(1,1-Dimethylethyl) phenoxy] propanoic acid	C_13_H_18_O_3_	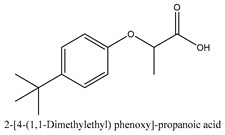	[6]
13	2-Camphanyl angelic acid ester	N/A	N/A	[4]
14	2-Methyl-2-β-butane-1-ol	C_5_H_12_O	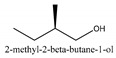	[5]
15	2-Methyl-5-(1-methylethyl)-2-cyclohexen-1-ketone	N/A	N/A	[4]
16	2-Methyl-5-isopropenyl-2-enyl ester acetic acid	N/A	N/A	[3]
17	2-Methyl-5-propene-2-cyclopentene-1-acetate	N/A	N/A	[5]
18	2-Phenyl-2-(phenylmethyl)-1,3-Dioxolane	C_16_H_16_O_2_	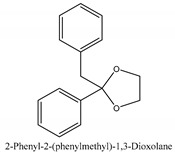	[6]
19	3,7-Dimethyl-3(*E*)-octene-1,2,6,7-tetraol	C_10_H_20_O_4_	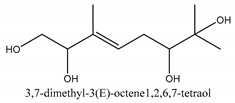	[7]
20	4-(1-Methylethyl)-benzyl alcohol	C_10_H_14_O	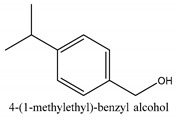	[4]
21	4-Methyl-l-(1-methylethyl)-2-cyclohexene-1-ol	C_10_H_18_O	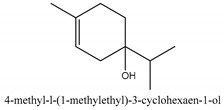	[3]
22	4-Methyl-l-(1-methylethyl)-3-cyclohexaen-1-ol	C_10_H_18_O	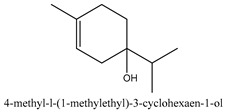	[8]
23	4-α-Hydroxydihydro-furan agar	N/A	N/A	[4]
24		5,5-Dimethyl-8-methylene-1,2-epoxycyclooct-3-ene	C_11_H_16_O	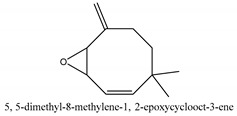	[3]
25	5-Isopropenyl-2-methyl-cyclohex-2-enyl propionic acid ester	N/A	N/A	[3]
26	5-Isopropenyl-3-methyl-cyclohex-1-enol	C_10_H_16_O	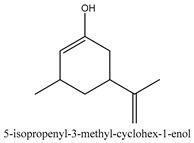	[3]
27	5-Propene-2-cyclopenten-1-ol	N/A	N/A	[5]
28	6,6-Dimethyl-2-bicyclo [3,1,1] heptane	N/A	N/A	[3]
29	6-Isopropenyl-3-methyl-cyclohex-1-enol	C_10_H_16_O	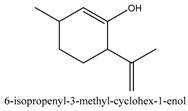	[3]
30	6-Isopropenyl-3-methyl-cyclohex-2-enol	C_10_H_16_O	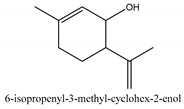	[3]
31	7*H*-Furo[3,2-g] [1] benzopyran-7-one,9-[(4-hydroxy-3-methyl-2-buten 1) oxy]-[*E*]	N/A	N/A	[9]
32	8-(3-methyl-2-β) hem	N/A	N/A	[5]
33	Cis-isopropenyl-2-methylene-3-cyclohexyl-acetate	N/A	N/A	[1]
34	Cnidimol E	C_15_H_16_O_6_	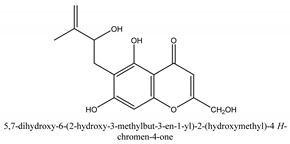	[10]
35	Decahydro-2α-methyl-6-methylene-1-(1-methylethyl) cyclobutyl [1,2,3,4] dicyclopentenyl	N/A	N/A	[11]
36	Dihydrocalamenene	N/A	N/A	[4]
37	Dl-umtatin	C_15_H_14_O_5_	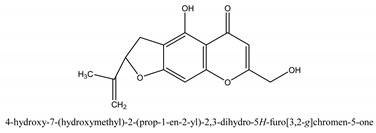	[12]
38	Fructus perillae aldehyde	N/A	N/A	[11]
39	Furfuryl heptadecenone	N/A	N/A	[4]
40	Geranyl 3-methyl-butyrate	C_15_H_26_O_2_	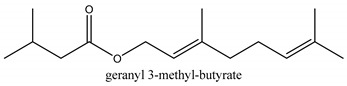	[13]
41	Hexahydrofarnesyl acetate	C_17_H_34_O_2_	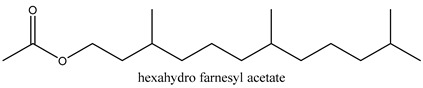	[13]
42	Isopropylisobutyric acid	C_7_H_14_O_2_	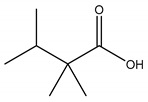	[5]
43		Pentanoate-1,3,3-trimethyl-bicyclo [2,2,1] hept-2-yl-ester	C_15_H_26_O_2_	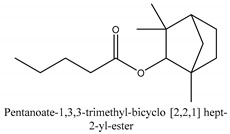	[4]
44	Phlojodicarpin	C_15_H_16_O_5_	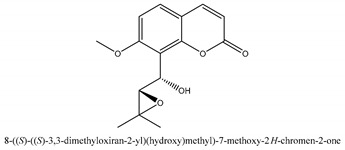	[2]
45	*p*-Mentha-*E*-2,8(9)-dien-1-ol	N/A	N/A	[8]
46	Propionic-2-methyl-1-methylethyl ester	C_7_H_14_O_2_	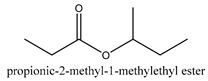	[4]
47	*trans*-Sabinene sesquihydrate	N/A	N/A	[4]
48	*trans*-Carvacryl acetate	C_12_H_16_O_2_	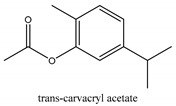	[4]
49	*trans*-Dihydrothujanyl alcohol	C_10_H_18_O	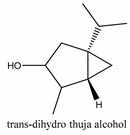	[4]
50	α-Cadien	N/A	N/A	[14]
51	α-Sung terpene	N/A	N/A	[7]
52	Chromones	2-Methyl-5-hydroxy-6-(2-butenyl-3-hydroxymethyl)-7-(β-d-glucopyranosyloxy)-4H-1-benzopyran-4-one	C_21_H_26_O_10_	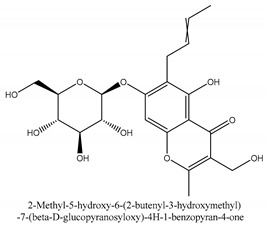	[15]
53	5-Hydroxychromone-7-*O*-β-d-glucoside	N/A	N/A	[16]
54	Cindimol F	C_15_H_14_O_6_	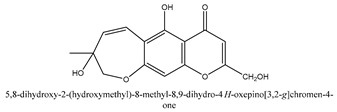	[17]
55	Cindimoside A	C_21_H_26_O_10_	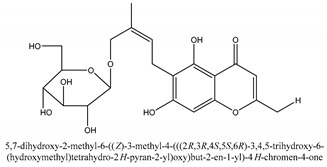	[18]
56	Eduotin IV	N/A	N/A	[19]
57	Monoterpenoid glucosides	(2*S*)-3,7-Dimethyloct-3(10),6-diene-1,2-diol 2-*O*-β-d-glucopyranoside	C_16_H_28_O_7_	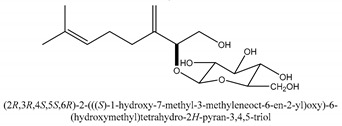	[20]
58	(4*S*)-*p*-Menth-1-ene-7,8-diol 8-*O*-β-d-glucopyranoside	C_16_H_28_O_7_	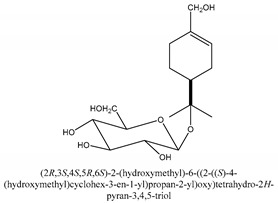	[20]
59	(6,7-erythro)-3,7-Dimethyloct-3(10)-ene-1,2,6,7,8-pentol	C_10_H_20_O_5_	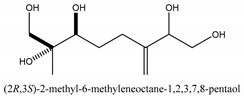	[21]
60	(6,7-threo)-3,7-Dimethyloct-3(10)-ene-1,2,6,7,8-pentol	C_10_H_20_O_5_	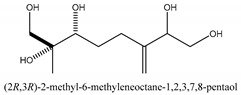	[21]
61	2-Methyl-5,7-dihydroxychromone 7-*O*-β-d-glucopyranoside	C_16_H_18_O_9_	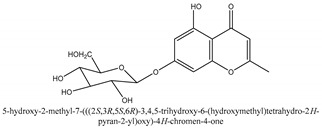	[20]
62	2-Triacetate	N/A	N/A	[22]
63	3,7-Dimethyl-1,2,6,7-tetrahydroxyoct-3(10)-ene	C_10_H_20_O_4_	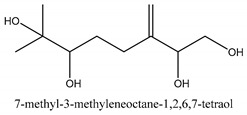	[22]
64	3,7-Dimethyl-3β,8-dihydroxy-oct-1,6-diene 3-*O*-β-d-glucopyranoside	N/A	N/A	[22]
65	3-Methyl-1,2,3,4-tetrahydroxybutane	C_5_H_12_O_4_	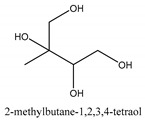	[22]
66	Enzymatic hydrolysis	N/A	N/A	[20]
67	Trans-p-menthane-1β,2α,8,9-tetrol	C_10_H_20_O_4_	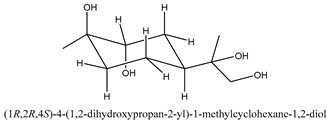	[21]
68	Xanthotoxol 8-*O*-β-d-glucopyranoside	C_17_H_16_O_9_	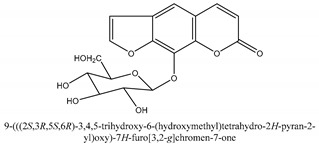	[20]
69	Terpenoids, Glycosides, Glucides	1-Hydroxyhorilin	C_23_H_34_O_6_	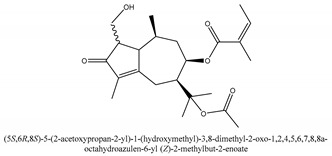	[23]
70	5-Methoxy-xanthotoxol-8-β-glucoside	C_14_H_10_O_7_	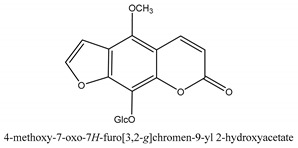	[24]
71	8-Methoxy-xanthotoxol-5-β-glucoside	C_14_H_10_O_7_	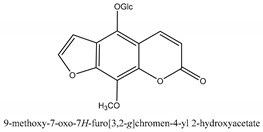	[24]
72	Cnideol BSynonyms:Cnideol B	C_12_H_12_O_5_	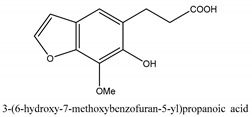	[25]
73	Cnidioside B	C_14_H_14_O_7_	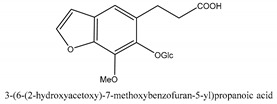	[25]
74	Cnidioside C	C_11_H_18_O_4_	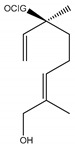	[25]
75	Glycerol-2-*O*-α-l-fucose galactoside	C_9_H_18_O_7_	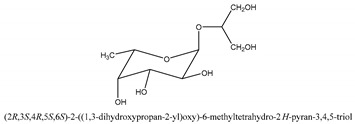	[26]
76	Methylpicraquassioside B	C_16_H_18_O_8_	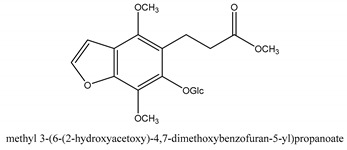	[24]
77	Picraquassioside B	C_15_H_16_O_8_	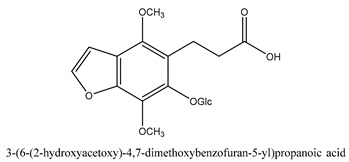	[24]
78	Xanthotoxol-8-β-glucoside	C_13_H_8_O_6_	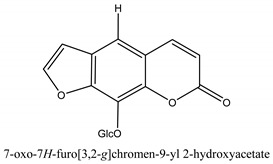	[24]
79	Other compounds	Cnideoside A	C_13_H_12_O_6_	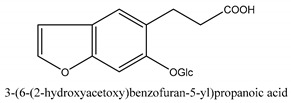	[27]
80	Cnideoside B	C_14_H_14_O_7_	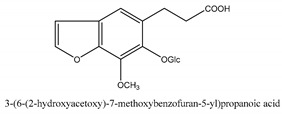	[27]

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
