# Peer review of "Phytochemistry, Ethnopharmacology, Pharmacokinetics and Toxicology of Cnidium monnieri (L.) Cusson"

_ijms, 2020, doi:10.3390/ijms21031006_

Round 1

Reviewer 1 Report

I have nothing to complain about the phytochemical and analytical parts of this paper, however it seems to me that the part on biological activities should at least be reorganized to gain clarity.

For example, it would seem more logical to group certain activities such as antimicrobial activities or anti-parasitic activities [Line 468: anti-bacterial] in the same paragraph with different sub-parts. Line 564: Anti-glioblastoma, please reorganize with anti-cancer activities.

Similarly, it would seem easier to group these activities according to types such as in vitro, in cellulo, animal assays, and clinical trials.

Author Response

Comment 1

it would seem more logical to group certain activities such as antimicrobial activities or anti-parasitic activities [Line 468: anti-bacterial] in the same paragraph with different sub-parts.

Response 1

Subheadings under section 3.2.5 have been used, including:

3.2.5.1 Anti-bacterial activity

3.2.5.2 Anti-fungal activity

3.2.5.3 Anti-parasitic activity

Comment 2

Line 564: Anti-glioblastoma, please reorganize with anti-cancer activities.

Response 2

The paragraph related to anti-glioblastoma effect has been reorganized as part of anti-cancer activities:

3.2.20.10 Anti-brain cancer activity

Comment 3

it would seem easier to group these activities according to types such as in vitro, in cellulo, animal assays, and clinical trials.

Response 3:

The types of the experiment (in vitro or in vivo) are presented in the Supplementary Table 2 for pharmacological effects and activities of CMC. As the classification of in vitro, in cellulo, animal assays, and clinical trials may cause some studies can be classified into multiple types, the types of the experiment in the Supplementary Table 2 have not been changed.

Reviewer 2 Report

Please read the report attached.

Author Response

Comment 1

The name of the plant should be corrected to Cnidium monnieri (L.) Cusson. Check the title and the text.

Response 1

The name of the plant has been corrected, in both the title and the text.

Comment 2

Section. Introduction: Several sentences are written in a non-suitable style.

Line 30. It belongs to the family of Apiaceae.

This sentence should be suppressed. The family should be inserted in the first sentence.

Response 2

The first sentence has been rewritten as below:

Cnidium monnieri (L.) Cusson. (CMC) is the dry fruit of Umbelliferae plant, Cnidium monnieri (L.) in the family of Apiaceae

Comment 3

Lines 31-32. As this herb is widely grown in China, Japan, Korea, Russia, Mongolia, and Vietnam. This sentence is incomplete!!

Response 3

This sentence has been rewritten as below:

As this herb is widely grown in China, Japan, Korea, Russia, Mongolia, and Vietnam, it is also known as “Jashoshi” in Japanese, “Sasangia” in Korean, and “Xa sang tu” in Vietnamese

Comment 4

Lines 35-36. It could strongly warm the kidney Yang, dry dampness, kill parasites, and relieve itchiness. Should be rewritten by using more suitable scientific-medical expressions.

Response 4

This sentence has been deleted from the manuscript.

Comment 5

Lines 41-42. As this review had been completed nearly five years ago. More researches had been done regarding CMC. Should be completely rewritten. (incomplete sentences, informal words “done”, …!!).

Response 5

This sentence has been rewritten as below:

Since this review was searched up to 2015, more research projects have been conducted regarding CMC since then. It is necessary to update the relevant knowledge in a timely manner.

Comment 6

Section. Materials and methods: Lines 47-49. References should be added regarding: Encyclopaedia of Traditional Chinese Medicines; China Network Knowledge Infrastructure.

Response 6

The references for these two databases have been added.

Comment 7

Line 66. Table 1 has a reference [429]??????? This reference does not exist. References’ number should be ordered according to their appearance in the text. Correct them in all tables.

Response 7

429 is not a reference but the total number of the chemical compounds. To avoid confusion, 429 has been removed from Table 1.

The chemical compounds references’ number is ordered according to their appearance in the text. But in the Supplementary table 1, they are categorized by different families. Therefore, some of the references’ number is not the same as the order of their appearance.

Comment 8

Section. Traditional uses: This section should be rewritten by using suitable scientific style. For example. “Kill parasites” should be replaced by anti-parasitic Recent references of ethnobotanical studies should be added rather than “talking” about few books without references.

Response 8

This paragraph has been deleted.

Comment 9

Section. Phytochemistry: Lines 104 – 119. This subsection should be rewritten by: Avoiding recalling the classification of coumarins and their structures, Presenting the main coumarins found in the plant. Do the same for the other subsections (4.1 to 4.8).

Response 9

All the subsections under Phytochemistry have been rewritten as suggested.

Comment 10

Lines 169-364. Analytical techniques: This subsection should be rewritten in one subsection without subtitles (methods). Avoid presenting the well-known principles of the techniques. Insert the main findings without redundancy. Avoid adding much known details. Avoid informal strange expressions such as “In professor Chen’s study,”

Response 10

The section on analytical techniques has been deleted.

Comment 11

Lines. 365-965. Pharmacological effects and activities: The majority of pharmacological activities given are those of osthole!!!! And not of the plant. This subsection should be enriched by adding more molecules or even crude extracts with biological activities. It’d be preferable to rewrite this subsection in more suitable style by reorganizing it according to the molecules tested and not diseases treated.

Response 11

The pharmacological effects section has been rewritten based on the molecules’ pharmacological effects and activities. This approach is consistent with that used in other reviews on pharmacology of plants.

Reviewer 3 Report

1. In the introduction it is not clear what part was already done in the previous  eviews and what part is done here. It is not justified why it is necessary to make this publication. What is the OBJETIVE ?
2. The way in which the material and method is described does not allow replicating the study in another species. It is not apropiate. It is not complete. It does not clarify. The databases consulted are correct. The Keywords provided are insufficient, since only those that refer to the name of the plant are indicated. The possible position of “” is not specified either. That determines the result of each search.
3. Table 1 does not make sense in Material and Method, in some way it is already a result. The same for Table 3.
4. Tables 1-3 are very long and should be transferred to Appendices or Supplementary Material.
5. Section 3. Botanical Characterization and distribution should be annulled as such, and its information included in 1. Introduction. When the first presentation of the plant is made, it is appropriate to describe its characteristics and expose its distribution area. Better with a map. An image of the species can also be included.
6. A Results section must be included, which then has several subsections: traditional uses, phytochemicals, etc.
7. Section 4. Traditional uses may be better if it is presented as a Table. The types of Uses should be exposed following a standardized or published Usage classification, such as the classification of Pardo et al (2013) Inventory of Traditional Knowledge related to Biodiversity. Ministry of the Environment, Madrid, Spain
8. Section 5. Phytochemistry, it does not make much sense to describe each of the large groups, (coumarins etc). Also in some cases it is defined in too much detail and in others (essential oils; which are complex mixtures), the information provided is very scarce. In this section, the Component Table should be taken from the information in Tables 1 and 3.
9. Section 6. Analytical techniques becomes very long, and it makes no sense to describe these techniques in specialized work such as this. If you want you can do a specific work on Techniques, to be published in a magazine whose scope is precisely this one.
10. Section 7 pharmacology, should group the subsections with a hierarchical classification, by organs or systems. Some classification can be used where the classification units and subunits have been perfectly defined, such as the one in Pardo (2013) Inventory of Traditional Knowledge related to Biodiversity. Ministry of Environment. Madrid Spain.
11. Table 4 can be summarized and since it is very long, it can also be passed to the Appendix or supplement. Much of the information indicated in section 7 can be written shorter or directly summarized in Table 4.
12. The pharmacodynamics section 8 is very short with respect to the extension of the rest of the sections and the interest or added value that it contributes is not very understood.
13. The Toxicology section can be included in the pharmacological actions, 7.
14. Section 10 does not provide clear added value, and can be deleted.
15. Section 11 discussion and conclusions cannot be considered sufficient because there are so many data that are provided in the results that the discussion seems almost impossible to be addressed in a single paper.
IN SUMMARY:
The work does not provide an answer to any specific question, nor does it specify a clear hypothesis. It is too long and I would need to present the information collected in a more fragmented way, in several successive and more summarized works, synthesizing the most important in Tables. With that structure, it is not useful, nor does it seem appropriate to be published in this journal

Author Response

Comment 1

In the introduction it is not clear what part was already done in the previous reviews and what part is done here. It is not justified why it is necessary to make this publication. What is the OBJETIVE?

Response 1

The justification and the objective of this study has been revised as below:

Since this review was searched up to 2015, more research projects have been conducted regarding CMC since then. It is necessary to update the relevant knowledge in a timely manner. This study aimed to provide an up-to-date review on the phytochemistry, ethnopharmacology, pharmacokinetics, and toxicology of CMC.

Comment  2

The way in which the material and method is described does not allow replicating the study in another species. It is not appropriate. It is not complete. It does not clarify. The databases consulted are correct. The Keywords provided are insufficient, since only those that refer to the name of the plant are indicated. The possible position of “” is not specified either. That determines the result of each search.

Response 2

The four different names of CMC were used for search in this review. The references of all the databases searched in this study have been provided for further clarification and replication of the current work.

Comment 3

Table 1 does not make sense in Material and Method, in some way it is already a result. The same for Table 3.

Response 3

“Table 1” and “Table 3” have been deleted from the Material and Method.

Comment 4

Tables 1-3 are very long and should be transferred to Appendices or Supplementary Material.

Response 4

Table 1 and Table 3 have been transferred to Supplementary Table 1 Chemical compounds isolated from CMC and Supplementary Table 2 Pharmacological effects and activities of CMC.

Table 2 has been changed to Table 1 Molecular formula and chemical structures of compounds derived from CMC; it is presented in Results section.

Comment 5

Section 3. Botanical Characterization and distribution should be annulled as such, and its information included in 1. Introduction. When the first presentation of the plant is made, it is appropriate to describe its characteristics and expose its distribution area. Better with a map. An image of the species can also be included.

Response 5

Botanical section has been deleted.

Figure 1 (A) and (B) has been added to address the characteristics and distribution as suggested.

Comment 6

A Results section must be included, which then has several subsections: traditional uses, phytochemicals, etc.

Response 6

The Results section has been added.

Comment 7

Section 4. Traditional uses may be better if it is presented as a Table. The types of Uses should be exposed following a standardized or published Usage classification, such as the classification of Pardo et al (2013) Inventory of Traditional Knowledge related to Biodiversity. Ministry of the Environment, Madrid, Spain

Response 7

The section on traditional use has been deleted.

We appreciate the reviewer recommended paper which helps us to improve our manuscript

Comment 8

Section 5. Phytochemistry, it does not make much sense to describe each of the large groups, (coumarins etc). Also, in some cases it is defined in too much detail and in others (essential oils; which are complex mixtures), the information provided is very scarce. In this section, the Component Table should be taken from the information in Tables 1 and 3.

Response 8

The general introduction of each main group of the herb is a common way to elaborate on the phytochemistry part of the research. The information for each group may not be the same in terms of quantity. This reflects the disparity of research focus for each group.  

Table 1 and Table 3 have been transferred to Supplementary Table 1. Some contents have been taken from these two tables and presented in the main text.

Comment  9

Section 6. Analytical techniques become very long, and it makes no sense to describe these techniques in specialized work such as this. If you want you can do a specific work on Techniques, to be published in a magazine whose scope is precisely this one.

Response 9

The section on analytical techniques has been deleted from this manuscript. We plan to prepare a separate paper for journal submission in the future as suggested.

Comment 10

Section 7 pharmacology should group the subsections with a hierarchical classification, by organs or systems. Some classification can be used where the classification units and subunits have been perfectly defined, such as the one in Pardo (2013) Inventory of Traditional Knowledge related to Biodiversity. Ministry of Environment. Madrid Spain.

Response 10

The section on pharmacological effects has been rewritten according to the molecules of the herb

Comment  11

Table 4 can be summarized and since it is very long, it can also be passed to the Appendix or supplement. Much of the information indicated in section 7 can be written shorter or directly summarized in Table 4.

Response 11

The information in Table 4 has been summarized in Supplementary Table 2.

The section on pharmacological effects has been rewritten (refer to response to Reviewer 2, Comment 11).

Comment 12

The pharmacodynamics section 8 is very short with respect to the extension of the rest of the sections and the interest or added value that it contributes is not very understood.

Response 12

The section on pharmacokinetics (under 3.3) has been rewritten.

Comment  13

The Toxicology section can be included in the pharmacological actions, 7.

Response 13

The section on Toxicology is included in 3.3 Pharmacokinetics and toxicology.

Comment 14

Section 10 does not provide clear added value and can be deleted.

Response 14

This section has been deleted as suggested

Round 2

Reviewer 3 Report

Although a remarkable effort has been made to improve the initial version of the paper, some suggestions that have not been addressed are still pending.

(Response of comment 4)
The Tables transferred to the Supplementary Material are still very long and cumbersome. They do not have a summary structure whose format allows you to easily find the information. Please redo the Supplementary Material Tables by improving the format and improving the visualization and synthesis of data. If appropriate, eliminate superfluous data to gain ease of understanding the Tables.

(Response of comment 5)

Please improve the quality of Figure1 (A). 

(Response to comment 7) 

Please consult https://www.miteco.gob.es/es/biodiversidad/temas/inventarios-nacionales/pbl_iect_tcm30-164090.pdf

(Response to comment 10) 

Please consult pgs. 34-41 and use the proposal of classification of medical uses based in Systems summarized in: 

 https://www.miteco.gob.es/es/biodiversidad/temas/inventarios-nacionales/pbl_iect_tcm30-164090.pdf

Author Response

Reviewer’s comments 4:

The Tables transferred to the Supplementary Material are still very long and cumbersome. They do not have a summary structure whose format allows you to easily find the information. Please redo the Supplementary Material Tables by improving the format and improving the visualization and synthesis of data. If appropriate, eliminate superfluous data to gain ease of understanding the Tables.

Response of comment 4

The table for chemical extraction information with references is in Supplement table 1 as separate file.

The table for pharmacological information is listed as Appendix the study appearance order is the same with their order in the text.

Reviewer’s comments 5:

Please improve the quality of Figure1 (A).

Response of comment 5

The quality of Figure 1 (A) is now improved.

Reviewer’s comments 7:

Section 4. Traditional uses may be better if it is presented as a Table. The types of Uses should be exposed following a standardized or published Usage classification, such as the classification of Pardo et al (2013) Inventory of Traditional Knowledge related to Biodiversity. Ministry of the Environment, Madrid, Spain

Response of comment 7:

Due to the timeframe for resubmission and word number limitation, this part is excluded. The authors appreciate the reviewer’s suggestion

Reviewer’s comments 10:

Please consult pgs. 34-41 and use the proposal of classification of medical uses based in Systems summarized in:

https://www.miteco.gob.es/es/biodiversidad/temas/inventarios-nacionales/pbl_iect_tcm30-164090.pdf

Response of comment 10:

The pharmacology section is now grouped by the systems of human disease as it dictated in the linked document. 

Round 3

Reviewer 3 Report

Checked that most of the suggested modifications have been made, with special attention to the most relevant ones, I consider that the work is acceptable to be published.